# Nonlinear Time Series Modeling Using Bernstein Polynomials and Bayesian Inference

## Abstract

Modeling and forecasting nonlinear time series presents a highly non-trivial challenge to both statistical and neural network methods. This is particularly true with time series from chaotic systems with very few methods being able to provide accurate predictions along with uncertainty quantification. This article taps into fundamental theoretical properties of Bernstein polynomials to model temporal dependence structures present in nonlinear time series. Unlike deep multilayer neural network forecasters, the BPAR retains a single structured layer, making the model comparatively simple while allowing time lag-specific Bernstein coefficient blocks to indicate which past observations are most responsible for prediction. Bayesian statistical inference is applied and this produces uncertainty quantification on model parameters and on predictions. Carefully designed shrinkage prior distributions are utilized in the Bayesian inference and subsequent posterior credible intervals assist in identifying the statistically significant model coefficients. An extensive empirical study involving several real and simulated nonlinear time series data is conducted along with suitable model residual diagnostic checks. These results demonstrate, especially with chaotic and highly nonlinear time series, superior forecasting performance of the BPAR relative to existing statistical and neural network methods. Results also indicate the usefulness of the new models in providing uncertainty quantification on model parameters and predictions.

## 1 Introduction

Nonlinear time series modeling must reconcile three goals that are often in tension: flexible approximation of nonlinear temporal dependence, model interpretability, and statistically meaningful uncertainty quantification with diagnosis. Classical statistical time series models such as ARIMA and SARIMA remain valuable because they are parsimonious and diagnostically transparent, but they are built around linear dependence after transformation and differencing (Box et al., 2015; Shumway & Stoffer, 2017). Related forecasting frameworks such as exponential smoothing, dynamic regression, vector autoregressions, and grouped time series methods remain firmly rooted in this statistical tradition (Hyndman & Athanasopoulos, 2021). Artificial neural network-based models offer far greater flexibility for nonlinear data, yet that flexibility is often purchased at the cost of black-box learned features, heavier tuning burdens, and less direct uncertainty interpretation. The difficulty is especially sharp in chaotic and strongly nonlinear systems (Lele, 1994; Mukherjee et al., 1997), where short-memory linear dependence structure can coexist with smooth nonlinear temporal dependence.

Recent machine learning work has pushed highly nonlinear time series forecasting in several important directions. Long Short-Term Memory (LSTM) networks were introduced to mitigate unstable or vanishing gradient flow in recurrent backpropagation (Rumelhart et al., 1986; Hochreiter & Schmidhuber, 1997), and recurrent autoregressive neural forecasters remain standard flexible baselines for nonlinear time series prediction (Salinas et al., 2017). Multilayer perceptron (MLP) forecasters use lagged observations as inputs to a feedforward network; their role as flexible nonlinear approximators is classical (Hornik et al., 1989), and MLP variants have also been studied in chaotic time series prediction problems (Stamatis et al., 1999). Basis-style residual networks and attention-based models have further expanded the neural forecasting toolkit through interpretable block decompositions and longer-range temporal modeling (Oreshkin et al., 2020; Lim et al.,

2021; Zhou et al., 2021; Wu et al., 2021). Spectral and Koopman-style methods provide expressive decompositions for nonlinear temporal dynamics (Lange et al., 2021). Generative forecasting models have also shown strong probabilistic performance on chaotic time series benchmarks, especially when the objective is designed for calibration rather than adversarial training (Pacchiardi et al., 2024). Recent studies further emphasize how training horizon, latent representation, and model family affect performance on canonical dynamical systems ranging from periodic to chaotic regimes (Aceituno et al., 2025; Kucukahmetler et al., 2026). Chaotic dynamical systems arising in settings such as astrophysics, biochemistry, and climatology, see Gilpin (2023a) for an example, provide a useful yet inherently challenging benchmark for comparing models that differ not only in predictive accuracy but also in interpretability (Gilpin, 2023b). Taken together, this literature demonstrates the strength of artificial neural network-based forecasters, but it also underscores persistent challenges: the fitted mechanisms are rarely interpretable at the time lag response level, hyperparameter tuning can be substantial, and calibrated uncertainty is usually tied to comparatively complex learning pipelines. With time series data, in addition to providing uncertainty quantification on predictions, being able to statistically test for nonlinearity in temporal autocorrelations, long memory and statistical significance of model parameters is of immense use in application areas such as economics and finance Eğrioğlu & Fildes (2022); Hauzenberger et al. (2024); Furuoka et al. (2024). These works are viewed as a hybrid between classical time series models and artificial neural network models and they employ techniques such as bootstrapping, Bayesian inference to conduct statistical tests.

The main idea of this article comes from Bernstein polynomials and their ability to approximate continuous functions defined on a compact interval (Lorentz, 1986). The newly proposed Bernstein Polynomial Autoregressive Model, abbreviated as BPAR, captures temporal dependence in nonlinear time series using these Bernstein polynomials. The resulting model architecture resembles a single-layer artificial neural network with a structured basis dictionary, yet the fitted effects remain directly interpretable lag by lag through the Bernstein coefficients. Compared to deep or multilayer neural network forecasters, this single structured layer avoids learned hidden representations and substantially reduces architectural choices such as the number of layers, hidden units, and activation functions. At the same time, each lag has its own Bernstein coefficient block, so posterior summaries of these blocks can indicate which past observations contribute most strongly to prediction. BPAR is particularly natural for chaotic time series in which the next observation is often a smooth nonlinear function of recent history. An additive extension, called AR-BPAR in this paper, augments the Bernstein model branch with a linear autoregressive component. This autoregressive component captures linear dependence, while the Bernstein branch captures nonlinear effects, and this is seen to be useful in modeling time series data with a mix of linear and nonlinear temporal dependence.

Two Bayesian inference strategies are studied in this paper. First, a variational Bayesian approach (Graves, 2011; Blundell et al., 2015) that learns model parameters without knowledge of the exact posterior distribution. Second, an exact Bayesian inference method wherein Gaussian priors along with an inverse-gamma variance prior leads to analytical expressions for the posterior distributions; Section 2.3 compares these two inference routes. The main idea under the Bayesian inference procedure is the design of time lag- and Bernstein polynomial degree-dependent shrinkage priors on the model's weight coefficients. Specifically, the prior mean of each Bernstein weight coefficient is zero, and the prior variance decays with respect to the time lag index and the polynomial degree index. Subsequent posterior credible intervals help trim down the statistically insignificant model coefficients.

The main contributions are summarized as follows.

- The new BPAR framework provides a highly flexible model class for modeling and forecasting a wide range of nonlinear time series. An extension to include a linear autoregressive model term further allows the model class to adapt to time series with mixed linear and nonlinear temporal dependence.

- Viewed as a hybrid between statistical time series models and artificial neural network models for nonlinear time series forecasting, the BPAR enables model interpretability and predictive uncertainty quantification via Bayesian inference. BPAR is better than classical statistical models such as ARIMA/SARIMA by accommodating highly nonlinear temporal dynamics. Simultaneously, its single-layer Bernstein polynomial structure avoids the typical hyperparameter selection, such as number of hidden layers/nodes and activation function, challenges faced by artificial neural network models, while retaining lag-level summaries of which past observations drive prediction.

- Time lag- and Bernstein polynomial degree-dependent shrinkage prior distributions are proposed for the Bernstein weight coefficients. Subsequent posterior credible intervals help prune the model by detecting the statistically significant coefficients.

- An extensive empirical study spanning simulated, chaotic, and nineteen (19) real-world datasets is conducted. Performance and model comparisons are done with a range of competitors starting from classical statistical models to modern deep learning based forecasting techniques.

## 2 Methodology

In this section, the Bernstein Polynomial Autoregressive Model (BPAR) and its additive autoregressive extension (AR-BPAR) are described. We start with the approximation theory on Bernstein polynomials, then introduce our new models, and finally outline the Bayesian inference procedures.

### 2.1 Approximation theory using Bernstein Polynomials

For $z \in [0, 1]$, the Bernstein basis polynomial of degree $m$ is written as

$$B_{k,m}(z) = \binom{m}{k} z^k (1 - z)^{m-k}, \qquad k = 0, \ldots, m.$$

The key property of interest is that Bernstein polynomials well approximate continuous functions on a compact interval. More precisely, let $f : [0, 1] \to \mathbb{R}$ be continuous. Its Bernstein polynomial approximation of degree $m$ is

$$(\mathcal{B}_m f)(z) = \sum_{k=0}^{m} f\left(\frac{k}{m}\right) B_{k,m}(z). \tag{1}$$

The Bernstein approximation theorem states that

$$\lim_{m \to \infty} (\mathcal{B}_m f)(z) = f(z)$$

uniformly on $[0, 1]$; see Feller (1966), Lorentz (1986), and Beals (2004). This result provides the mathematical motivation and foundation for representing the nonlinear impact of past values of a time series on its future.

To visualize this approximation theorem, consider the continuous function $g(z) = 3z + 3\sin(4\pi z) + 2\cos(5\pi z)$, $z \in [0, 1]$. Figure 1 plots approximations of function $g(\cdot)$ with Bernstein polynomials at degrees $m \in \{5, 10, 20, 40, 80, 160, 320\}$. The low-degree curves smooth out local oscillations, while the high-degree curves nearly overlap the true function. It is precisely this property that is used to build our BPAR model in Section 2.2 and beyond. Specifically, each time-lagged observation of the time series is modeled to influence its future observations by a continuous function that is then approximated by Bernstein polynomials.

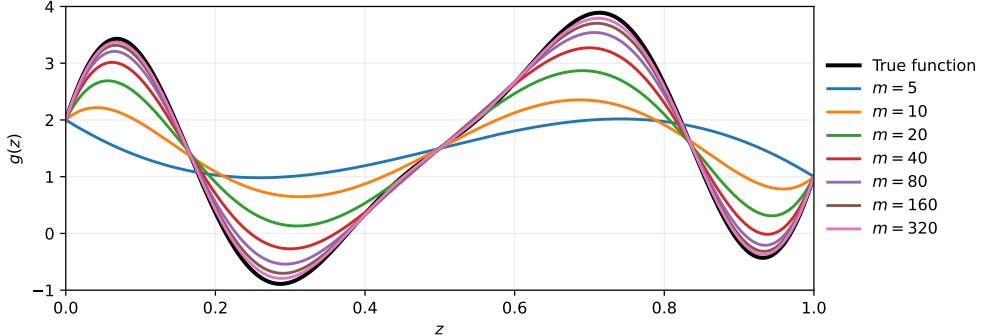

Figure 1: Approximations of the function $g(z) = 3z + 3\sin(4\pi z) + 2\cos(5\pi z)$ on $[0, 1]$ at varying Bernstein polynomial degree $m$.

## 2.2 Bernstein Polynomial Autoregression (BPAR)

Let $x_t$ be the observed nonlinear time series. The BPAR model of order $d_B$ is written as

$$x_t = \alpha + \beta \sum_{j=1}^{d_B} \sum_{k=0}^{m} w_{j,k} B_{k,m}(z_{t-j}) + \varepsilon_t, \ \ \varepsilon_t \sim \mathcal{N}(0, \sigma^2), \tag{2}$$

where the coefficients $\alpha, \beta \in \mathbb{R}$, and $m$ is the Bernstein polynomial degree. We denote this model by BPAR($d_B$). The injective map $h : \mathbb{R} \to [0, 1]$ yields $z_{t-j} = h(x_{t-j})$, and the nonlinear temporal autocorrelations are captured via the Bernstein polynomial representation using the sequence of random variables $\{z_t\}$. Note that with function $h(\cdot)$ being a deterministic injective transform of the time series $x_t$ onto the interval $[0, 1]$, $z_t = h(x_t)$ is also viewed as a sequence of random variables. In practice, we work with two choices for $h$. First, the min–max scaling $h(x) = (x - \min_x)/(\max_x - \min_x)$, where $\max_x$ and $\min_x$ are assumed to be known maximum and minimum values attained by the time series. Second, the logistic-sigmoid function $h(x) = 1/(1 + e^{-x})$. In our empirical studies, we observed very comparable forecasting performance with these two choices. The sequence of random variables $\epsilon_t$ denotes the error component.

The BPAR($d_B$) model's parameter vector is written as $\theta = (\alpha, \beta, \mathbf{w}^\top)^\top$, with the Bernstein weight coefficients $\mathbf{w} = (w_{1,0}, \ldots, w_{1,m}, \ldots, w_{d_B,0}, \ldots, w_{d_B,m})^\top$. We then define

$$\mathbf{b}_m(z) = [B_{0,m}(z), \ldots, B_{m,m}(z)]^\top, \ \ \phi_t = \left[\mathbf{b}_m(z_{t-1})^\top, \ldots, \mathbf{b}_m(z_{t-d_B})^\top\right]^\top.$$

Then, the BPAR($d_B$) model from (2) can be expressed as

$$\mu_t(\theta) = \alpha + \beta \phi_t^\top \mathbf{w}, \quad x_t = \mu_t(\theta) + \varepsilon_t. \tag{3}$$

The BPAR model architecture is illustrated in Figure 2. This is seen to resemble a single-layer artificial neural network with the Bernstein polynomials playing the role of pseudo-activation functions. This single-layer representation is simpler than deep multilayer neural network forecasters because the nonlinear transformations are supplied by a fixed Bernstein basis rather than by several learned hidden layers. The coefficient block $\mathbf{w}_j = (w_{j,0}, \ldots, w_{j,m})^\top$ controls the nonlinear contribution of time lag $j$, hence posterior means and credible intervals for $\mathbf{w}_j$ provide a direct way to assess which time lags drive predictions. With such a structure in place, the BPAR caters to Bayesian statistical inference and this is described in Section 2.3. This inferential method enables uncertainty quantification regarding the model parameters and also on the time series' predictions. Our extensive empirical study in Section 3 shows superior forecasting performance of the BPAR model over classical linear time series models when the time series data is highly nonlinear, for example chaotic time series. Further, BPAR's performance is seen to be on par or better than its artificial neural network competitors such as LSTM and MLP.

**Remark 2.1.** Prediction intervals using nonlinear autoregressive models have been developed and studied in the statistical literature; see Franke et al. (2002); Pan & Politis (2016); Wu & Politis (2024) for examples. The model assumption in these works is that $x_t = m(x_{t-1}, x_{t-2}, \ldots, m_{t-p}) + \epsilon_t$, for an order $p$ model wherein $m(\cdot)$ is a possibly nonlinear function. If $m(\cdot)$ is assumed to be smooth, the problem then becomes a nonparametric autoregressive model with $m(\cdot)$ being estimated using kernel methods. In practice, when implementing their methods, $m(\cdot)$ is estimated using kernels or assumed to be known. Our BPAR($d_B$) model in (2) is more flexible than the above described setup as it models the nonlinear influence of each time-lagged version of the time series via the $z_t$ random variables in (2). As evidenced by the results in our empirical study in Section 3, this previous point is seen to be true when dealing with chaotic time series. Further, the Bernstein polynomials help avoid the difficulty in selecting the unknown $m(\cdot)$ function.

**Remark 2.2.** Bernstein polynomials have been used to model the trajectory of time series; see Lukoseviciute et al. (2018) for an example. There, the Bernstein polynomial is an interpolating curve in the time index and forecasts are obtained by extrapolating that curve. The extrapolating equation there points to modeling linear temporal autocorrelations. In contrast, our BPAR setup models nonlinear temporal autocorrelations using Bernstein polynomials. Specifically, the nonlinear influence of each time-lagged observation on the

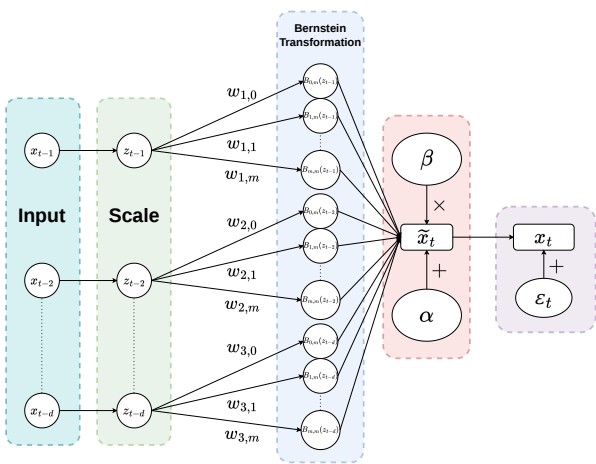

Figure 2: BPAR($d_B$) model architecture.

current state of the time series is modeled via Bernstein polynomials. This allows more flexibility to capture highly nonlinear temporal dependence structures, as often seen in chaotic time series.

**Remark 2.3** The proposed BPAR/AR-BPAR model has structural resemblance to the nonparametric regression model. Bernstein polynomials have been used in a number of nonparametric regression related works; see Curtis & Ghosh (2011); Osman & Ghosh (2012); Wang & Ghosh (2012). There, a response variable $Y_i$ at observation index $i$ is expressed as $Y_i = g(X_i) + \text{error}$, where $X_i$ is the explanatory variable. The function $g(\cdot)$ is then assumed to be continuous and can be approximated via Bernstein polynomials. Osman & Ghosh (2012) discuss certain key advantages of using Bernstein polynomials. First, statistical consistency properties have been established with reasonable rates of convergence. Second, it has been shown that in order to establish these theoretical guarantees, high-order smoothness assumptions are not required, but mere continuity is adequate. Third, Bernstein polynomials cater well to shape restrictions in the regression function such as monotone, convex/concave, nonnegative or combinations of these. Though beyond the scope of this current work, these existing theoretical results can be extended to the time series setting that we have with the BPAR model, under certain regularity assumptions on the time series.

### 2.2.1 Extension to AR-BPAR

The BPAR model from Section 2.2 is very well suited for highly nonlinear or chaotic time series data. There are many examples of real time series data displaying a mix of linear and nonlinear temporal autocorrelations, with some examples wherein the linear effects are much more prominent than the nonlinear effects. To accommodate such types of time series data, we present the AR($d_L$)-BPAR($d_B$) model below which extends the BPAR($d_B$) model. Specifically, we have

$$x_t = \alpha + \sum_{i=1}^{d_L} \rho_i x_{t-i} + \sum_{j=1}^{d_B} \sum_{k=0}^{m} w_{j,k} B_{k,m}(z_{t-j}) + \varepsilon_t, \ \ \varepsilon_t \sim \mathcal{N}(0, \sigma^2), \tag{4}$$

where $d_L$ is the linear autoregressive model order. The AR($d_L$)-BPAR($d_B$) can also be written as

$$x_t = \mathbf{f}_t^\top \theta + \varepsilon_t, \tag{5}$$
$$\theta = (\alpha, \rho_1, \ldots, \rho_{d_L}, w_{1,0}, \ldots, w_{1,m}, \ldots, w_{d_B,0}, \ldots, w_{d_B,m})^\top,$$
$$\mathbf{f}_t = [1, x_{t-1}, \ldots, x_{t-d_L}, B_{0,m}(z_{t-1}), \ldots, B_{m,m}(z_{t-d_B})]^\top.$$

Figure 3 shows the architecture of the AR($d_L$)-BPAR($d_B$) model. The empirical study in Section 3 provides evidence on the usefulness of this extension when dealing with time series data that include a mix of linear and nonlinear temporal dependence.

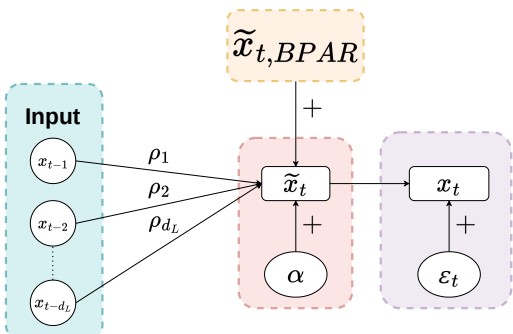

Figure 3: AR($d_L$)-BPAR($d_B$) model architecture. Note that $\tilde{x}_{t,BPAR}$ is the $\tilde{x}$ from the BPAR($d_B$) architecture in Figure 2.

### 2.3 Bayesian Inference

Let $\theta$ denote the vector of model parameters and let $K_\theta = \dim(\theta)$ denote its size. For the BPAR($d_B$) model from Section 2.2, $\theta = (\alpha, \beta, \mathbf{w}^\top)^\top$ and $K_\theta = 2 + d_B(m+1)$. Here, the $d_B(m+1)$-dimensional vector $\mathbf{w}$ denotes the Bernstein weight coefficients; see Figure 2. For AR($d_L$)-BPAR($d_B$) from Section 2.2.1, $\theta = (\alpha, \rho^\top, \mathbf{w}^\top)^\top$ with $K_\theta = 1 + d_L + d_B(m+1)$. Here, $\rho = (\rho_1, \rho_2, \ldots, \rho_{d_L})$ denotes the coefficients corresponding to the linear autoregressive part of the model; see Figure 3. Before detailing the two inference routes, we state their scope precisely. The variational inference derivation (ELBO with the reparameterization trick) and the conjugate Gaussian–inverse-gamma updates are standard results, which we apply rather than derive. The main contribution under the Bayesian inferential framework is the design of the shrinkage priors on certain model parameters. These prior distributions involve zero mean and their variances decay with respect to the time lag index and the Bernstein polynomial degree index. This design gives principled, automatic control of model size $K_\theta$ as model pruning can be done via inspection of posterior credible intervals. We now detail two inference approaches, namely variational Bayesian inference and exact Bayesian inference.

#### 2.3.1 Bayesian variational inference

Bayesian variational inference is a popular estimation method used in machine learning models wherein the exact posterior distributions become intractable Hoffman et al. (2013); Blei et al. (2017). Another notable advantage is the avoidance of MCMC algorithms to draw samples from the posterior distributions as this can become computationally expensive. With the BPAR($d_B$) model from Section 2.2, the Gaussian prior distribution specification on the parameter vector $\theta$ is written as

$$p(\theta) = \mathcal{N}(m_0, V_0), \tag{6}$$

where $V_0$ is the prior diagonal covariance matrix and $m_0$ is the prior mean vector. The intercept parameter $\alpha$ and the scale parameter $\beta$ are assigned fixed weakly informative prior variances, while the Bernstein weights are assigned lag- and Bernstein degree-adaptive shrinkage variances. More explicitly,

$$m_0 = \left(m_{0,\alpha}, m_{0,\beta}, \mathbf{0}_{d_B(m+1)}^\top\right)^\top, \qquad V_0 = \mathrm{diag}\left(\tau_\alpha^2, \tau_\beta^2, v_{1,0}^{\mathrm{B}}, \ldots, v_{d_B,m}^{\mathrm{B}}\right),$$

with

$$v_{j,k}^{\mathrm{B}} = \mathrm{Var}(w_{j,k}) = \frac{\tau_{\mathrm{B}}^2}{j^{\gamma_{\mathrm{B},\ell}}(k+1)^{\gamma_{\mathrm{B},m}}}, \qquad j = 1, \ldots, d_B, \quad k = 0, \ldots, m.$$

Here $\tau_\alpha^2$ and $\tau_\beta^2$ do not depend on lag or the Bernstein polynomial degree $m$, and they control only the global location and scale of the nonlinear response. In contrast, $v_{j,k}^{\mathrm{B}}$ decreases with lag index $j$ and Bernstein basis index $k$ when $\gamma_{\mathrm{B},\ell}$ or $\gamma_{\mathrm{B},m}$ are positive. Therefore the shrinkage mechanism here penalizes selecting a high model order $d_B$ and/or high Bernstein polynomial degree $m$. To simplify technical exposition, we explain

the variational inference method under the Gaussian prior family. Note that with the variational inference framework other prior probability distributions can also be used.

The posterior distribution is approximated by a multivariate Normal distribution that is written as

$$q(\theta) = \mathcal{N}(m_q, \Sigma_q), \qquad \Sigma_q = L_q L_q^\top, \tag{7}$$

where $m_q$ is a learnable mean vector, and $L_q$ is a learnable lower-triangular Cholesky factor. Sampling from this distribution is implemented through the reparameterization

$$\theta = m_q + L_q \xi, \qquad \xi \sim \mathcal{N}(0, I), \tag{8}$$

so that stochastic gradients can be propagated through Monte Carlo draws of the parameters. Next, the Kullback–Leibler divergence between the prior and posterior distributions is expressed as

$$\mathrm{KL}(q(\theta) \,\|\, p(\theta)) = \frac{1}{2} \left[ \mathrm{tr}\big(V_0^{-1} \Sigma_q\big) + (m_q - m_0)^\top V_0^{-1} (m_q - m_0) - K_\theta + \log \frac{\det V_0}{\det \Sigma_q} \right], \tag{9}$$

where $K_\theta$ is the dimension of $\theta$. The training objective function is the negative evidence lower bound given by

$$\mathcal{L}_{\mathrm{VI}} = -\mathbb{E}_{q(\theta)} \left[ \log p(\mathbf{x} \mid \theta, \sigma^2) \right] + \lambda \, \mathrm{KL}(q(\theta) \,\|\, p(\theta)), \tag{10}$$

where $\lambda$ is the KL-weight used in training. The gradients of (10) are obtained via backpropagation; see (Rumelhart et al., 1986; Goodfellow et al., 2016) for details. After minimizing the objective function in (10), the optimized variational mean and covariance are denoted by $\hat{m}_q$ and $\hat{\Sigma}_q$, respectively. Thus, for any particular model parameter $\theta_j$, the posterior marginal distribution can be written as

$$q(\theta_j) \approx \mathcal{N}(\hat{m}_{q,j}, \hat{\Sigma}_{q,jj}),$$

where $\hat{m}_{q,j}$ and $\hat{\Sigma}_{q,jj}$ are estimated by the backpropagation procedure used to minimize (10). This then yields an approximate $100(1 - \alpha)\%$ posterior credible interval as

$$\theta_j \in \hat{m}_{q,j} \pm z_{1-\alpha/2} \sqrt{\hat{\Sigma}_{q,jj}}.$$

### 2.3.2 Exact Bayesian Inference for AR-BPAR

With the $\mathrm{AR}(d_L)$-$\mathrm{BPAR}(d_B)$ model from Section 2.2.1, a sample size of $n$ and row-aligning $f_t$ from (5) across the $t$ index produces a design matrix $H \in \mathbb{R}^{n \times K_\theta}$. The model can be then rewritten as

$$\mathbf{x} = H\theta + \varepsilon, \qquad \varepsilon \sim \mathcal{N}(0, \sigma^2 I_n). \tag{11}$$

The conjugate prior specification here is

$$\theta \mid \sigma^2 \sim \mathcal{N}(m_0, \sigma^2 V_0), \tag{12}$$

$$\sigma^2 \sim \mathrm{InvGamma}(a_0, b_0), \tag{13}$$

where $V_0$ is a diagonal matrix, and

$$m_0 = \left(m_{0,\alpha}, \mathbf{0}_{d_L}^\top, \mathbf{0}_{d_B(m+1)}^\top\right)^\top, \qquad V_0 = \mathrm{diag}\left(\tau_\alpha^2, v_1^{\mathrm{AR}}, \ldots, v_{d_L}^{\mathrm{AR}}, v_{1,0}^{\mathrm{B}}, \ldots, v_{d_B,m}^{\mathrm{B}}\right),$$

where

$$v_i^{\mathrm{AR}} = \frac{\tau_{\mathrm{AR}}^2}{i^{\gamma_{\mathrm{AR}}}}, \qquad v_{j,k}^{\mathrm{B}} = \frac{\tau_{\mathrm{B}}^2}{j^{\gamma_{\mathrm{B},\ell}} (k+1)^{\gamma_{\mathrm{B},m}}}.$$

Equivalently,

$$\mathrm{Var}(\alpha \mid \sigma^2) = \sigma^2 \tau_\alpha^2, \qquad \mathrm{Var}(\rho_i \mid \sigma^2) = \sigma^2 v_i^{\mathrm{AR}}, \qquad \mathrm{Var}(w_{j,k} \mid \sigma^2) = \sigma^2 v_{j,k}^{\mathrm{B}}.$$

In other words, the above hierarchy means that the intercept $\alpha$ receives a fixed weakly informative prior variance, while the AR coefficients and Bernstein weights can be shrunk by their lag orders $(d_L, d_B)$ and, for the Bernstein block, also by the polynomial degree $m$.

To derive the posterior distribution in closed form, the Gaussian data likelihood is written as

$$p(\mathbf{x} \mid \theta, \sigma^2, H) \propto (\sigma^2)^{-n/2} \exp\left\{-\frac{1}{2\sigma^2}(\mathbf{x} - H\theta)^\top(\mathbf{x} - H\theta)\right\}, \tag{14}$$

and the Gaussian prior density is

$$p(\theta \mid \sigma^2) \propto (\sigma^2)^{-K_\theta/2} \exp\left\{-\frac{1}{2\sigma^2}(\theta - m_0)^\top V_0^{-1}(\theta - m_0)\right\}. \tag{15}$$

Conditional on $\sigma^2$, the posterior density of $\theta$ is proportional to the product of (14) and (15). The quadratic part involving $\theta$ is

$$\begin{aligned} Q(\theta) &= (\mathbf{x} - H\theta)^\top(\mathbf{x} - H\theta) + (\theta - m_0)^\top V_0^{-1}(\theta - m_0) \\ &= \theta^\top(H^\top H + V_0^{-1})\theta - 2\theta^\top(H^\top\mathbf{x} + V_0^{-1}m_0) + \mathbf{x}^\top\mathbf{x} + m_0^\top V_0^{-1}m_0. \end{aligned} \tag{16}$$

Let

$$V_n = (V_0^{-1} + H^\top H)^{-1}, \qquad m_n = V_n(V_0^{-1}m_0 + H^\top\mathbf{x}).$$

Then $V_n^{-1}m_n = V_0^{-1}m_0 + H^\top\mathbf{x}$, and completing the square gives

$$Q(\theta) = (\theta - m_n)^\top V_n^{-1}(\theta - m_n) + \mathbf{x}^\top\mathbf{x} + m_0^\top V_0^{-1}m_0 - m_n^\top V_n^{-1}m_n. \tag{17}$$

The final three terms do not depend on $\theta$. Combining this completed square with the inverse-gamma prior for $\sigma^2$ gives the posterior updates

$$V_n = (V_0^{-1} + H^\top H)^{-1}, \tag{18}$$

$$m_n = V_n(V_0^{-1}m_0 + H^\top\mathbf{x}), \tag{19}$$

$$a_n = a_0 + \frac{n}{2}, \tag{20}$$

$$b_n = b_0 + \frac{1}{2}\left(\mathbf{x}^\top\mathbf{x} + m_0^\top V_0^{-1}m_0 - m_n^\top V_n^{-1}m_n\right). \tag{21}$$

Hence,

$$\theta \mid \sigma^2, \mathbf{x} \sim \mathcal{N}(m_n, \sigma^2 V_n), \qquad \sigma^2 \mid \mathbf{x} \sim \text{InvGamma}(a_n, b_n). \tag{22}$$

Parameter uncertainty is quantified using the posterior distribution in (22). Marginally, after integrating out $\sigma^2$, each model parameter satisfies

$$\theta_j \mid \mathbf{x} \sim t_{2a_n}\left(m_{n,j}, \frac{b_n}{a_n}(V_n)_{jj}\right),$$

where the second argument is again written in variance form. Hence, a Bayesian $100(1-\alpha)\%$ posterior credible interval for $\theta_j$ is

$$m_{n,j} \pm t_{2a_n,\, 1-\alpha/2}\sqrt{\frac{b_n}{a_n}(V_n)_{jj}}.$$

The Student-$t$ posterior predictive distribution follows by integrating over both $\theta$ and $\sigma^2$. For a new feature vector $\mathbf{f}_*$ and future observation $x_*$,

$$x_* \mid \theta, \sigma^2, \mathbf{x} \sim \mathcal{N}(\mathbf{f}_*^\top\theta, \sigma^2).$$

Integrating out $\theta \mid \sigma^2, \mathbf{x} \sim \mathcal{N}(m_n, \sigma^2 V_n)$ gives

$$x_* \mid \sigma^2, \mathbf{x} \sim \mathcal{N}\left(\mathbf{f}_*^\top m_n, \sigma^2\left(1 + \mathbf{f}_*^\top V_n\mathbf{f}_*\right)\right). \tag{23}$$

The factor $\mathbf{f}_*^\top V_n \mathbf{f}_*$ is the posterior coefficient-uncertainty contribution, while the leading 1 is new observation noise. Finally, integrating out $\sigma^2 \mid \mathbf{x} \sim \mathrm{InvGamma}(a_n, b_n)$ gives

$$x_* \mid \mathbf{x} \sim t_{2a_n}\left(\mathbf{f}_*^\top m_n, \frac{b_n}{a_n}\left(1 + \mathbf{f}_*^\top V_n \mathbf{f}_*\right)\right), \tag{24}$$

where the first argument is the degrees of freedom and the second scale argument is written in variance form. Therefore, the posterior predictive mean is $\mathbf{f}_*^\top m_n$ when $a_n > 1/2$, and the predictive variance is

$$\frac{b_n}{a_n - 1}\left(1 + \mathbf{f}_*^\top V_n \mathbf{f}_*\right), \qquad a_n > 1.$$

The main advantage of this exact Bayesian estimation route is that posterior shrinkage, parameter uncertainty, and predictive uncertainty are all obtained in closed form without the iterative optimization procedure done in the variational inference in Section 2.3.1.

## 2.4 Model hyperparameter selection

With the BPAR($d_B$) model from Section 2.2, let $\eta_B$ denote a hyperparameter configuration and this can be written as

$$\eta_{\mathrm{B}} = (d_B, m, \tau_{\mathrm{B}}, \gamma_{\mathrm{B}}, \lambda, \eta_{\mathrm{opt}}),$$

where $m$ is the Bernstein polynomial degree, $\tau_{\mathrm{B}}$ and $\gamma_{\mathrm{B}}$ are the shrinkage parameters, $\lambda$ is the variational KL weight, and $\eta_{\mathrm{opt}}$ collects optimizer settings such as learning rate and weight decay. With the AR($d_L$)-BPAR($d_B$) model from Section 2.2.1, the hyperparameter configuration has the form

$$\eta_{\mathrm{AB}} = (d_L, d_B, m, \tau_{\mathrm{AR}}, \tau_{\mathrm{B}}, \gamma_{\mathrm{AR}}, \gamma_{\mathrm{B}}),$$

where $\{\tau_{\mathrm{AR}}, \gamma_{\mathrm{AR}}\}$ are the shrinkage parameters corresponding to the linear autoregressive part of the model, $\{\tau_{\mathrm{B}}, \gamma_{\mathrm{B}}\}$ are the shrinkage parameters corresponding to the nonlinear part of the model.

Let $\mathcal{G}$ be the finite grid of candidate configurations. For each $\eta \in \mathcal{G}$, the model is fit on the training split and evaluated on the validation split. The validation error is

$$R_{\mathrm{val}}(\eta) = \left[\frac{1}{|\mathcal{V}|}\sum_{t \in \mathcal{V}}\left(x_t - \hat{x}_t^{(\eta)}\right)^2\right]^{1/2}, \tag{25}$$

where $\mathcal{V}$ is the validation index set. Residual temporal autocorrelations are summarized during grid selection by the validation Ljung–Box test $p$-value denoted by $P_{\mathrm{LB}}(\eta)$. Recall that the Ljung–Box test has a null hypothesis that the residuals are a white noise time series without any remaining temporal correlations. Smaller $R_{\mathrm{val}}(\eta)$ indicates better validation accuracy, while larger $P_{\mathrm{LB}}(\eta)$ indicates weaker evidence of residual temporal autocorrelations.

The implemented hyperparameter selection rule gives first priority to the condition in (25), and then second priority to the Ljung–Box p-values $P_{\mathrm{LB}}(\eta)$. With a small choice for $\delta$, we define the near-best hyperparameter selection set as

$$\mathcal{G}_\delta = \left\{\eta \in \mathcal{G} : R_{\mathrm{val}}(\eta) \leq (1 + \delta)\min_{\eta' \in \mathcal{G}} R_{\mathrm{val}}(\eta')\right\}.$$

The selected hyperparameter configuration is then defined as

$$\hat{\eta} = \arg\max_{\eta \in \mathcal{G}_\delta} P_{\mathrm{LB}}(\eta). \tag{26}$$

Another way to understand our procedure above is that the final hyperparameter configuration choice never sacrifices more than a fraction $\delta$ of the best validation error, and it prefers the near-best configuration with the most satisfactory residuals.

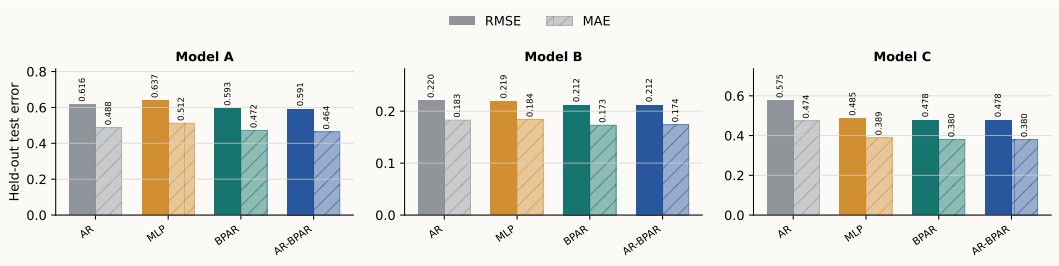

Figure 4: Held-out test RMSE and MAE for Models A–C. The competing methods are AR, MLP, BPAR, and AR-BPAR.

## 3 Empirical study

We test the performance of the new methods over a wide range of simulated, benchmark and real time series datasets. Models are trained and tuned on the training and validation sets, and the main paper reports held-out test set performance comparisons. Forecast accuracy is summarized by root mean squared error (RMSE) and mean absolute error (MAE). Residual Ljung–Box diagnostics are used as a secondary check for remaining temporal dependence (Ljung & Box, 1978). Lower RMSE and MAE are better, while larger Ljung–Box $p$-values indicate weaker signal of leftover autocorrelation. As competitors we consider the linear autoregressive model often abbreviated as AR($p$), its extension SARIMA that includes trend and seasonal terms, and two artificial neural network models: the MLP and LSTM. For the injective map $h$ used in the model definitions in Sections 2.2 and 2.2.1, we explored two choices: min–max scaling based on the observations range and a logistic sigmoid function. These two choices yielded very similar results, and the reported results are based on the first choice. The empirical study includes nonlinear time series simulation schemes, a few well-known chaotic time series examples, and two real nonlinear time series. All datasets use chronological train/validation/test splits. Models A–C in Section 3.1.1 use 180/60/60 observations split. In Section 3.1.2 with chaotic time series benchmarks, the Ricker, Logistic, and Gauss maps use 300/100/100 observations as the split, and the Mackey–Glass uses a 1400/300/300 split. In Section 3.2 with the two real datasets, we use a 720/144/144 observations split. The main paper reports held-out test forecasting results, while Appendix A provides a more detailed report on each dataset. Specifically, the appendix includes time series plots of the train/validation/test splits, tables with precise RMSE, MAE, and Ljung–Box $p$-values, forecast figures with posterior predictive intervals, and full train/validation/test comparison tables.

### 3.1 Simulated and chaotic benchmark datasets

### 3.1.1 Simulation schemes

The first set of experiments uses three nonlinear autoregressive simulation schemes adapted from the bootstrap prediction study of Wu & Politis (2024). These three schemes, called Models A–C, are given below.

$$
\begin{aligned}
\text{Model A:} \quad & x_t = 0.1x_{t-1}I(x_{t-1} \leq 0) + 0.8x_{t-1}I(x_{t-1} > 0) + \varepsilon_t, \\
\text{Model B:} \quad & x_t = \{0.1x_{t-1} + 0.5e^{-x_{t-1}^2}\varepsilon_t\}I(x_{t-1} \leq 0) + \{0.8x_{t-1} + 0.5e^{-x_{t-1}^2}\varepsilon_t\}I(x_{t-1} > 0), \\
\text{Model C:} \quad & x_t = 0.2 + \log(0.5 + |x_{t-1}|) + \varepsilon_t,
\end{aligned}
\tag{27}
$$

where $\varepsilon_t \sim \mathcal{N}(0, 0.5^2)$. Figure 4 shows the held-out test RMSE and MAE comparison across competing methods in each panel. The AR($d_L$)-BPAR($d_B$) model reveals slightly better performance than its competitors, with the difference being most evident in Model C. Table 1 in Appendix A.1 reports the held-out test RMSE, MAE, and residual Ljung–Box $p$-values for all methods on Models A–C. Appendix A.1 reports the full results, and we witness comparable performance between all methods for Models A and B, but better results under the new AR($d_L$)-BPAR($d_B$) model for Model C. The mixed results can be attributed to the extent of nonlinearity present in the time series, with Model C being the most nonlinear. The next section includes examples of highly nonlinear time series, also called chaotic time series.

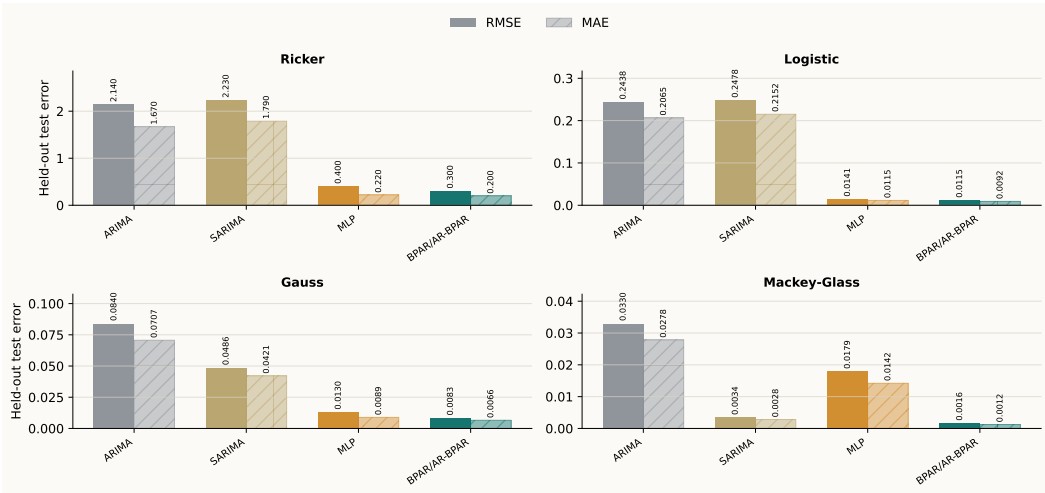

Figure 5: Held-out test RMSE and MAE summaries for the chaotic time series benchmark datasets. Each panel compares the common set of methods available for all four chaotic benchmarks: ARIMA, SARIMA, MLP, and BPAR/AR-BPAR.

### 3.1.2 Chaotic time series benchmarks

Multiple real world processes arising from energy systems, atmospheric dynamics and financial markets are viewed as nonlinear dynamical systems Kui & Lai (2025). The time series data from these applications are highly nonlinear and present a difficult forecasting challenge. To evaluate the performance of our new methods, we consider four chaotic time series benchmarks from `Python`'s `dysts` library Gilpin (2023a). We now provide details of the four time series along with the forecast comparison.

**Ricker map:** $x_{t+1} = x_t \exp(a - x_t)$,

**Logistic map:** $x_{t+1} = r x_t (1 - x_t)$, $r = 3.9$,

**Gauss map:** $x_{t+1} = \exp(-a x_t^2) + b$, $a = 4.9$, $b = -0.5$,

**Mackey–Glass delay system:** $dx(t)/dt = \beta x(t - \tau)/(1 + x(t - \tau)^\eta) - \gamma x(t)$, $\beta = 0.2$, $\gamma = 0.1$, $\eta = 10$, $\tau = 17$.

The Ricker panel in Figure 5 shows that our new BPAR model is the best performing method by test RMSE and test MAE, within this comparison set of classical statistical models and compact neural networks. Note that Section 3.2 and Appendix B contain more comparisons, especially to modern deep learning based forecasters. Appendix A.2.1 provides more detailed results on RMSE, MAE, Ljung–Box residual $p$-values, the training/validation/test split plot of the time series, forecast comparison, and other performance related summaries. A similar story is seen with the Logistic map, Gauss map, and Mackey–Glass delay system; see Figure 5. Additional method-specific results, including LSTM where it was run, remain in the appendix tables. More detailed performance results on the Logistic and Gauss maps can be found in Appendix A.2.2, and results on the Mackey–Glass delay system can be found in Appendix A.2.3.

Observe that the chaotic time series benchmarks chosen here are more nonlinear than the simulation schemes discussed in Section 3.1.1. A central feature in chaotic time series models is that the influence of the past on the future is assumed to be smooth; see Lele (1994); Mukherjee et al. (1997) for a more technical view of chaotic time series. Our proposed BPAR model from Section 2.2 is tailored for such situations as Bernstein polynomials well approximate bounded continuous functions. This is resulting in the superior forecast performance of the BPAR model seen in this section. A key advantage of the proposed method is that unlike much of existing statistical literature on nonlinear time series models, see Tiao & Tsay (1994); Pan & Politis (2016); Wu & Politis (2024) for examples, our modeling and forecasting approach does not require knowledge of the functional form that defines how the past influences the future of the time series. While

artificial neural networks have been utilized for forecasting such highly nonlinear time series, see Kui & Lai (2025) for a recent example, our approach enables model tractability and provides prediction uncertainty quantification.

### 3.2 Real-world datasets

We first consider two real time series data examples that have, in the past, been modeled and predicted as nonlinear time series. The first is the Ontario electricity price series, drawn from public reports maintained by the Independent Electricity System Operator (IESO)[1] (Independent Electricity System Operator, 2026). Ontario wholesale electricity prices are volatile, spike-prone, and have previously been modeled using nonlinear autoregressive neural networks and neural network price-spike forecasting methods (Andalib & Atry, 2009; Sandhu et al., 2016). The second is the Beijing $PM_{2.5}$ pollution series from the UCI Machine Learning Repository[2] (Chen, 2015). This hourly dataset records $PM_{2.5}$ measurements from the U.S. Embassy in Beijing together with meteorological covariates from Beijing Capital International Airport during 2010–2014. Beijing $PM_{2.5}$ has also been studied as a nonlinear and volatile forecasting problem using deep learning and hybrid time-series models (Yang et al., 2021; Zhao et al., 2022).

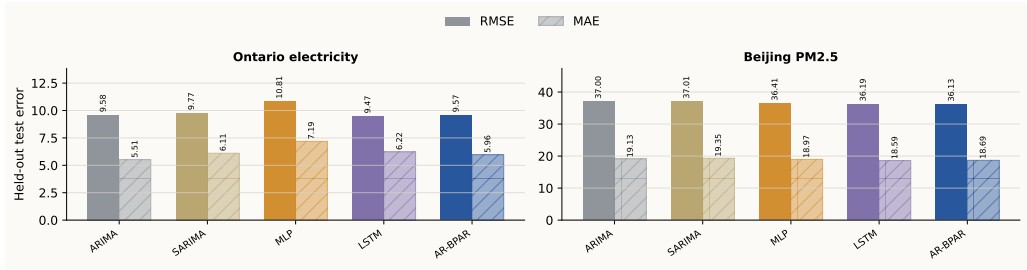

Figure 6: Held-out test RMSE and MAE summaries for the real-data benchmarks. Both panels use the same five methods: ARIMA, SARIMA, MLP, LSTM, and AR-BPAR.

In terms of forecasting performance, for the Ontario electricity prices time series, Figure 6 shows that the LSTM and our new AR-BPAR model are the two best performing models with both showing comparable results. A similar story is seen with the Beijing $PM_{2.5}$ time series. Detailed performance results on these two real time series datasets can be found in Appendix A.3. Appendix A.5 builds on the posterior credible intervals of the Bernstein weight coefficients, and detects for the presence/absence of nonlinear temporal autocorrelations in these two time series.

Beyond these results, we also consider seventeen (17) other real-world datasets spanning areas such as physics, ecology, hydrology, energy, environmental science, meteorology, transportation, astronomy and biology. Among the competing methods listed in Figure 7, the classical statistical baseline suite includes techniques such as ridge-regularized autoregression (ridge AR) Hoerl & Kennard (1970) and exponential smoothing (ETS) Hyndman & Athanasopoulos (2021). Among more modern deep learning based forecasters, we consider N-BEATS (Oreshkin et al., 2020), DeepAR (Salinas et al., 2017), a compact Transformer (Vaswani et al., 2017), and N-HiTS (Challu et al., 2023). We also list capacity-matched MLP and LSTM variants, whose model sizes are enlarged so their trainable parameter count matches or exceeds the size of the selected AR-BPAR configuration. Next, the quality of the prediction intervals is assessed with three standard measures: the prediction interval coverage probability (PICP) measuring the proportion of test observations that fall inside the nominal 90% or 95% interval; the mean prediction interval width (MPIW), which measures sharpness and is reported relative to the test-target standard deviation (Khosravi et al., 2011); the continuous ranked probability score (CRPS), a proper scoring rule that rewards calibration and sharpness jointly (Matheson & Winkler, 1976; Gneiting & Raftery, 2007). Next, every method's trainable-parameter count is reported, so that accuracy comparisons can be read against model size. Formal definitions of PICP, MPIW, CRPS, and the effective-parameter measure are given in Appendix B.1.

---

[1] https://www.ieso.ca/en/power-data/data-directory

[2] https://archive.ics.uci.edu/dataset/381/beijing+pm2+5+data

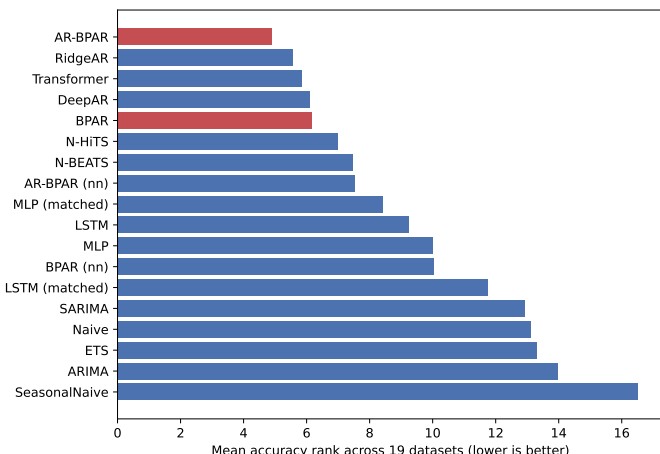

Figure 7: Mean test-RMSE rank of competing methods across the 19 real-world datasets. Lower is better, and the newly proposed models are in red.

The key takeaway of this empirical study is consistency: individual competitors trade places from dataset to dataset, whereas our new models (BPAR and AR-BPAR) are best-or-near-best across much of the board. It is seen that AR-BPAR attains the best mean accuracy rank of all 18 compared methods while providing calibrated prediction intervals in closed form. With test RMSE as the metric, Figure 7 shows the average accuracy rank of every method across 19 real-world datasets. The more modern deep learning forecasters can attain lower point forecast error rates on certain datasets but at the cost of two-to-three orders of magnitude more model parameters; Appendix B.2 provides the empirical evidence on the model sizes. Complete results on the performance of posterior credible intervals, model parameter counts, and hyperparameter/prior sensitivity analysis are collected in Appendix B.

## 4 Conclusion

This article introduces a new class of nonlinear time series models by tapping into fundamental properties of Bernstein polynomials. The newly proposed BPAR, and its extension to AR-BPAR, modeling framework are highly flexible in modeling a wide variety of nonlinear time series data. They serve as a useful statistical middle ground between rigid linear time series models and fully black-box neural network forecasters. Bayesian inference enables uncertainty quantification on model parameters and also on the predictions. Shrinkage prior distributions are placed on certain model parameters that control the overall size of the model parameter vector. Posterior credible intervals on each model parameter makes model pruning possible. With similarities seen between the new BPAR and a single-layer artificial neural network, the foundational work in this article can lead to extensions involving additional hidden layers in the BPAR model architecture. In such situations, the variational Bayesian inference approach can be utilized to quantify prediction uncertainty.

Results from extensive empirical experiments point to the usefulness of the new models. Specifically with highly nonlinear time series data such as chaotic time series, the new BPAR model substantially outperforms classical statistical models, and it outperforms neural network competitors (MLP and LSTM). Modern deep learning based forecasters can attain lower point forecast error with certain datasets, but this comes at the cost of two-to-three orders of magnitude more model parameters. Across 19 real-world datasets, AR-BPAR is seen to attain, on average, the best test-RMSE among the competing methods. Overall, the empirical findings make a strong case for the new nonlinear time series modeling framework using Bernstein polynomials, especially when time lag-level interpretation and uncertainty quantification are central scientific goals.

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

# A    Complete empirical study results

This appendix collects the dataset-specific details omitted from the main text. The results are organized in the same order for the three empirical blocks: a held-out test prediction table, a compact held-out test prediction figure with posterior prediction intervals where available, and then the detailed train/validation/test displays and comparison tables. The purpose is to preserve the complete experimental record while keeping the main paper focused on held-out test-set comparisons.

## A.1    Nonlinear simulation models

Table 1: Held-out test set prediction comparison for Models A–C. LB refers to the residual Ljung–Box test $p$-value.

| Model | Method | Test RMSE | Test MAE | LB |
|---|---|---|---|---|
| A | AR | 0.616 | 0.488 | 0.01 |
| A | MLP | 0.637 | 0.512 | 0.00 |
| A | BPAR | 0.593 | 0.472 | 0.02 |
| A | AR-BPAR | 0.591 | 0.464 | 0.02 |
| B | AR | 0.220 | 0.183 | 0.31 |
| B | MLP | 0.219 | 0.184 | 0.32 |
| B | BPAR | 0.212 | 0.173 | 0.21 |
| B | AR-BPAR | 0.212 | 0.174 | 0.22 |
| C | AR | 0.575 | 0.474 | 0.90 |
| C | MLP | 0.485 | 0.389 | 0.63 |
| C | BPAR | 0.478 | 0.380 | 0.61 |
| C | AR-BPAR | 0.478 | 0.380 | 0.61 |

The simulation formulas are given in Section 3; this appendix keeps the supporting numerical results grouped with the corresponding prediction displays.

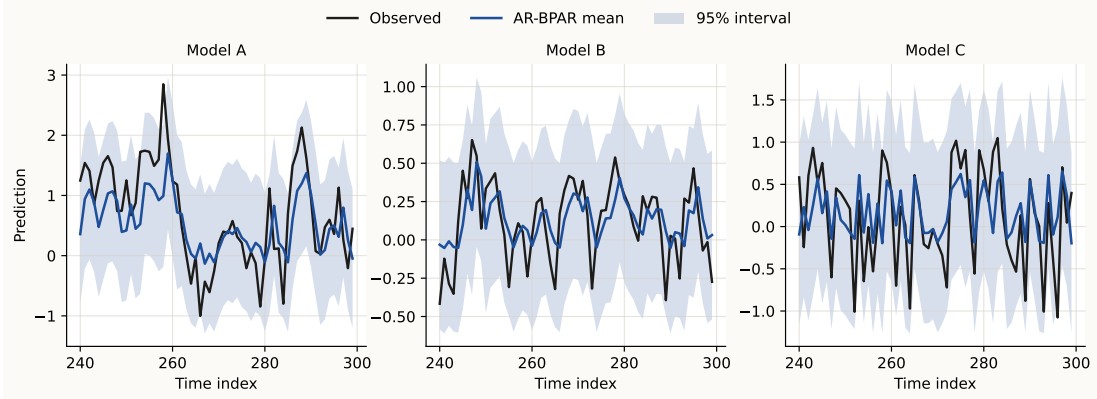

Figure 8: Held-out test set predictions for Models A–C using the selected BPAR/AR-BPAR specification. Panels are ordered left-to-right as Models A, B, and C. Each panel shows the observed test series, the BPAR/AR-BPAR posterior mean, and the 95% posterior prediction interval.

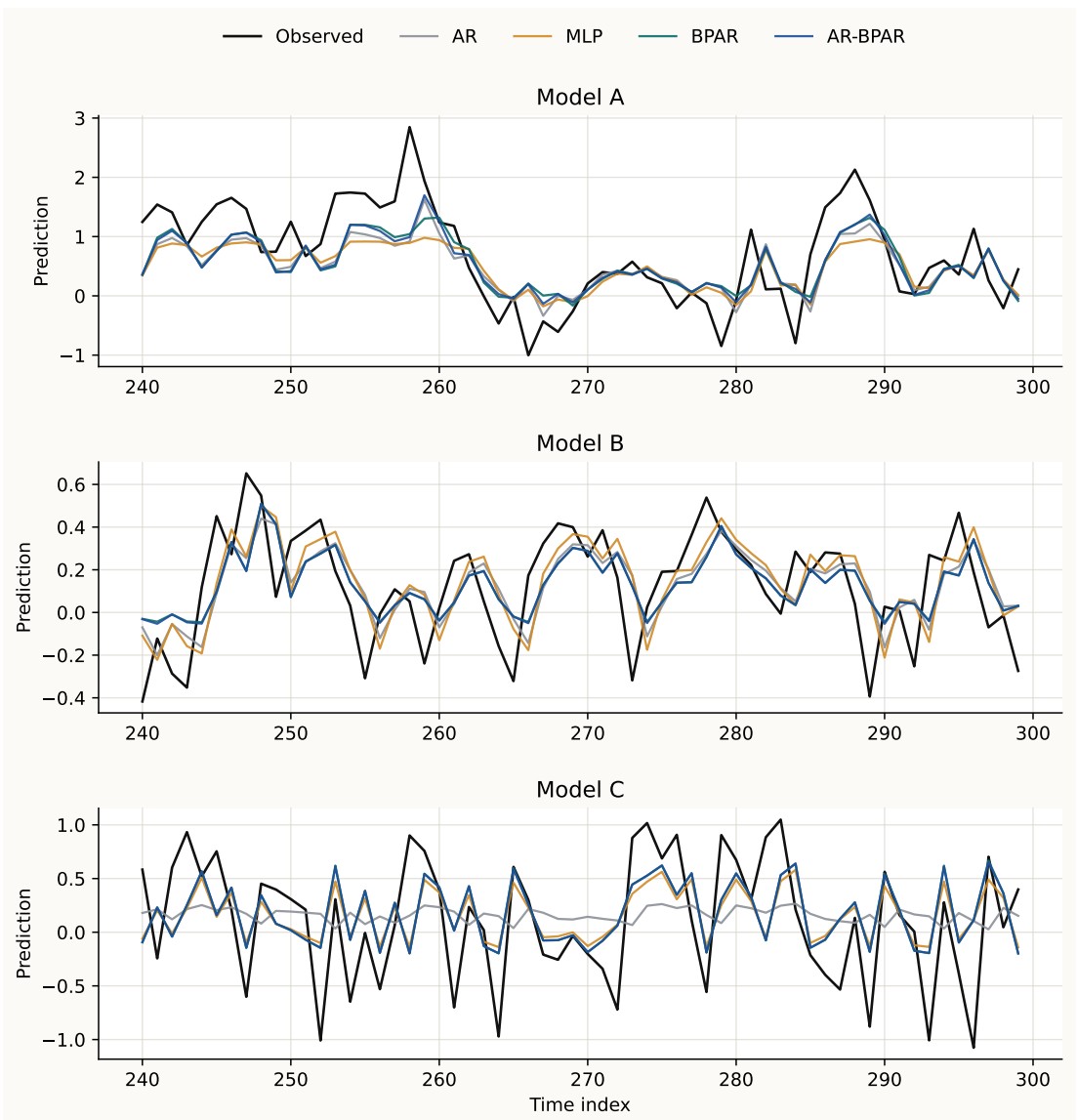

Figure 9: Held-out test set prediction comparison for Models A–C.

## A.2 Chaotic time series benchmarks

### A.2.1 Ricker map

Table 2: Held-out test set prediction comparison for the chaotic time series benchmarks. LB refers to the residual Ljung–Box test $p$-value.

| Benchmark | Method | Test RMSE | Test MAE | LB |
|---|---|---|---|---|
| Ricker | ARIMA | 2.14 | 1.67 | 0.03 |
| Ricker | SARIMA | 2.23 | 1.79 | 0.00 |
| Ricker | MLP | 0.40 | 0.22 | 0.06 |
| Ricker | BPAR | 0.30 | 0.20 | 0.22 |
| Logistic | ARIMA | 0.2438 | 0.2065 | 0.32 |
| Logistic | SARIMA | 0.2478 | 0.2152 | 0.09 |
| Logistic | MLP | 0.0141 | 0.0115 | 0.62 |
| Logistic | LSTM | 0.0174 | 0.0143 | 0.08 |
| Logistic | BPAR | 0.0115 | 0.0092 | $3.6 \times 10^{-4}$ |
| Gauss | ARIMA | 0.0840 | 0.0707 | $6.1 \times 10^{-20}$ |
| Gauss | SARIMA | 0.0486 | 0.0421 | 0.43 |
| Gauss | MLP | 0.0130 | 0.0089 | 0.03 |
| Gauss | BPAR | 0.0083 | 0.0066 | $2.2 \times 10^{-5}$ |
| Mackey–Glass | AR-BPAR (Bayesian) | 0.00162 | 0.00123 | $3.03 \times 10^{-59}$ |
| Mackey–Glass | AR-BPAR (Neural) | 0.00619 | 0.00455 | $2.09 \times 10^{-66}$ |
| Mackey–Glass | ARIMA | 0.03297 | 0.02781 | $5.08 \times 10^{-201}$ |
| Mackey–Glass | SARIMA | 0.00343 | 0.00280 | $2.85 \times 10^{-82}$ |
| Mackey–Glass | MLP | 0.01790 | 0.01422 | $7.18 \times 10^{-96}$ |
| Mackey–Glass | LSTM | 0.01811 | 0.01528 | $1.06 \times 10^{-98}$ |

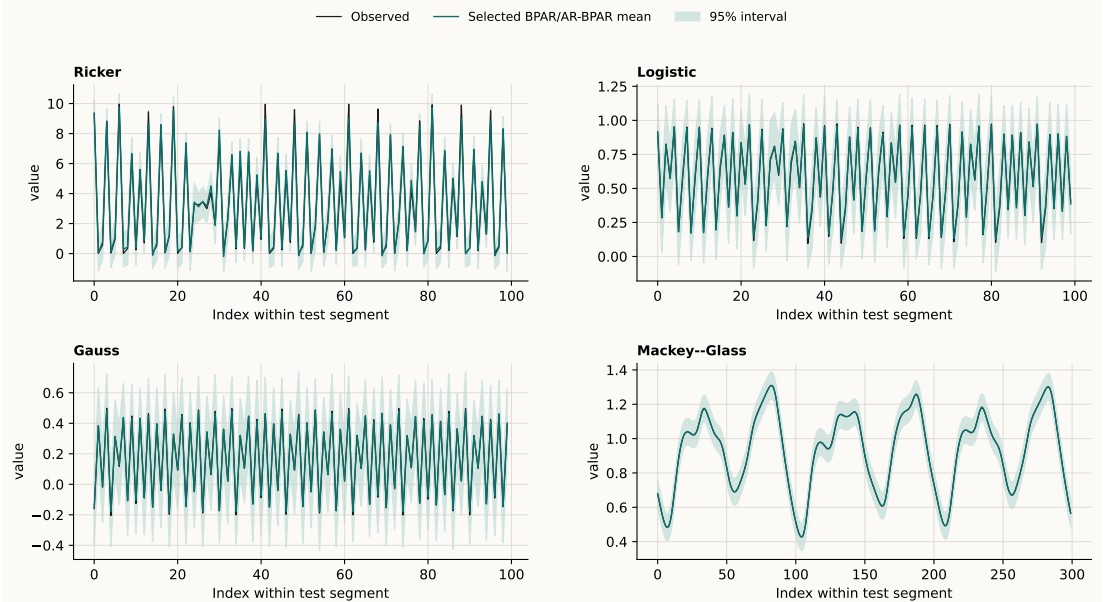

Figure 10: Held-out test set predictions for the chaotic time series benchmarks. Panels are ordered left-to-right, top-to-bottom as Ricker, Logistic, Gauss, and Mackey–Glass. Each panel shows the observed test set series, the selected BPAR/AR-BPAR posterior mean, and the 95% posterior prediction interval.

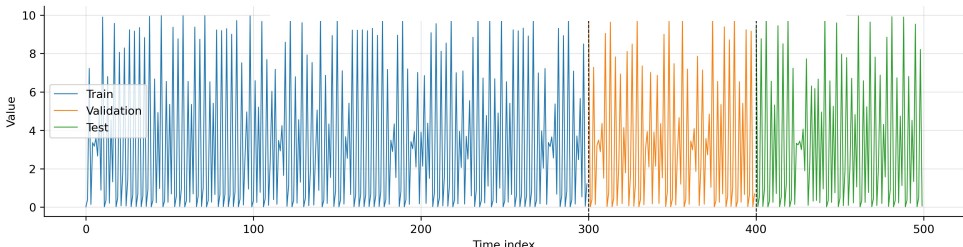

Figure 11: Ricker map time series with indications (vertical dashed lines) of the chronological train/validation/test split.

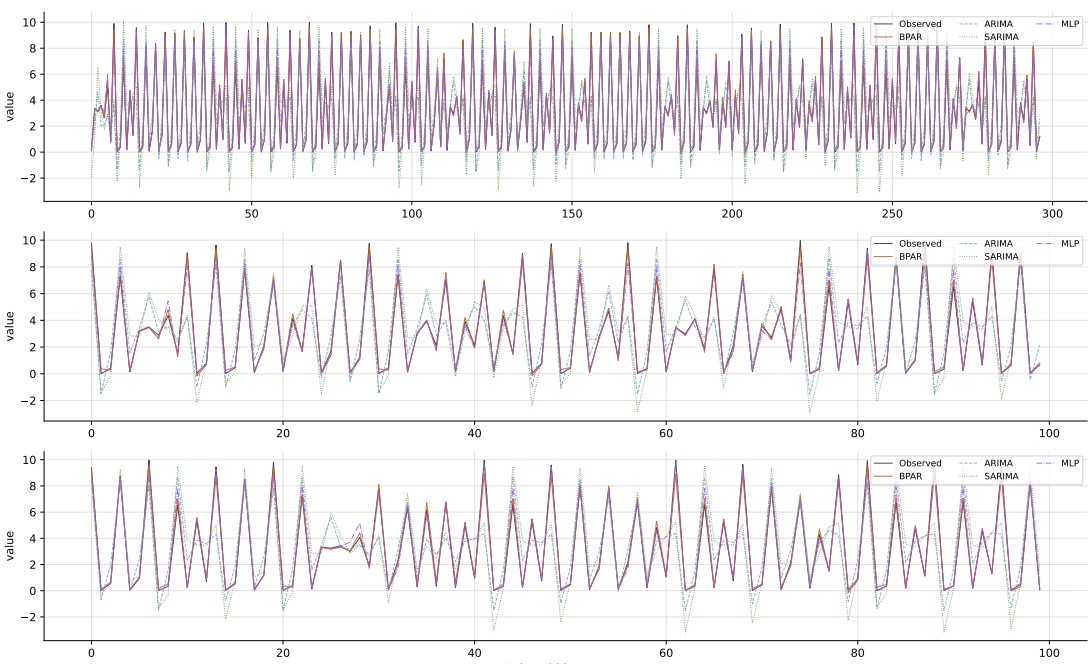

Figure 12: Train, validation, and test set predictions comparison for the Ricker map time series.

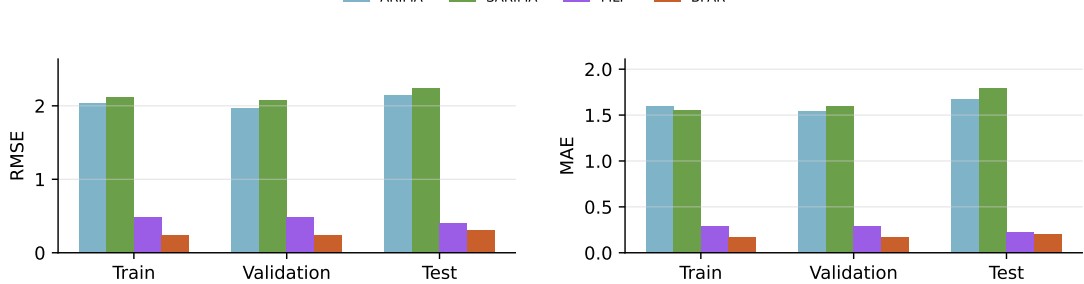

Figure 13: RMSE and MAE comparison across the train, validation, and test splits for the Ricker map time series.

Table 3: RMSE and MAE comparison for the Ricker map time series across train, validation, and test splits.

| Split | ARIMA RMSE | SARIMA RMSE | MLP RMSE | BPAR RMSE | ARIMA MAE | SARIMA MAE | MLP MAE | BPAR MAE |
|---|---|---|---|---|---|---|---|---|
| Train | 2.035675 | 2.112786 | 0.479972 | 0.233081 | 1.600501 | 1.549507 | 0.284966 | 0.164067 |
| Validation | 1.962163 | 2.069753 | 0.486570 | 0.231865 | 1.540349 | 1.598570 | 0.287815 | 0.173453 |
| Test | 2.137523 | 2.234414 | 0.403244 | 0.299752 | 1.667278 | 1.788471 | 0.223882 | 0.202838 |

### A.2.2 Logistic and Gauss maps

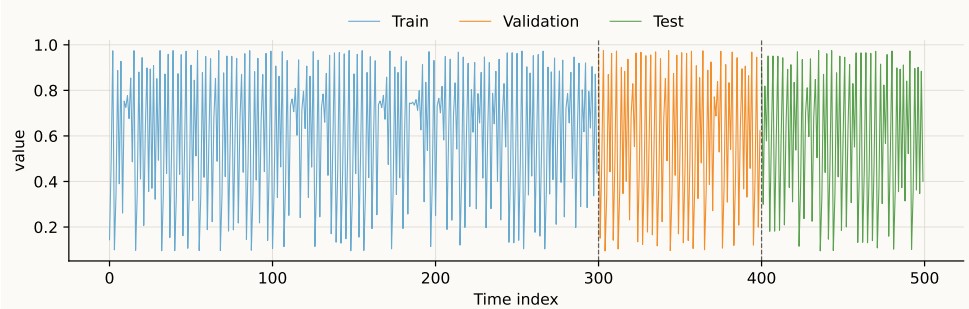

Figure 14: Logistic map time series with indications (vertical dashed lines) of the chronological train/validation/test split.

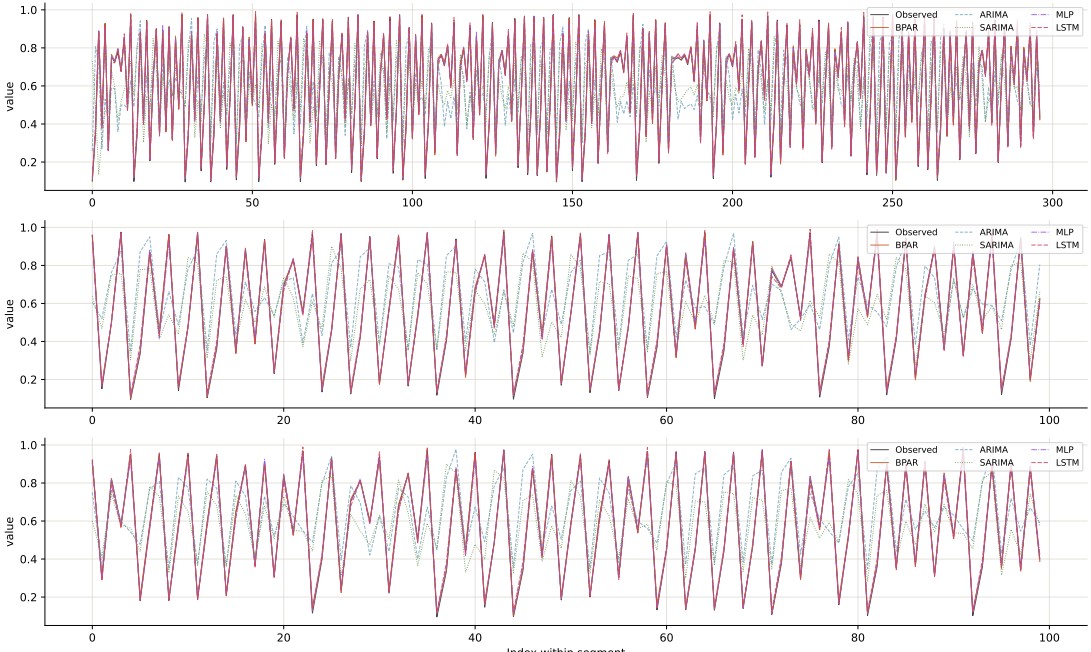

Figure 15: Train, validation, and test set predictions comparison for the Logistic map time series.

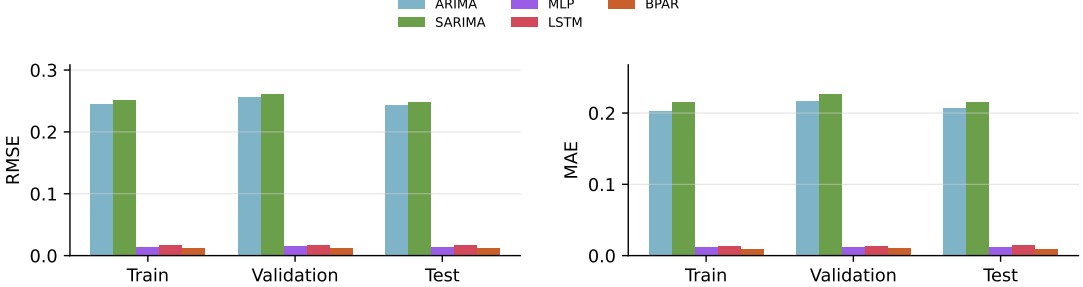

Figure 16: RMSE and MAE comparison across the train, validation, and test splits for the Logistic map benchmark.

Table 4: RMSE and MAE comparison for the Logistic map time series across train, validation, and test splits.

| Split | ARIMA RMSE | SARIMA RMSE | MLP RMSE | LSTM RMSE | BPAR RMSE | ARIMA MAE | SARIMA MAE | MLP MAE | LSTM MAE | BPAR MAE |
|---|---|---|---|---|---|---|---|---|---|---|
| Train | 0.244600 | 0.251541 | 0.014209 | 0.017371 | 0.011690 | 0.202602 | 0.215852 | 0.011462 | 0.013876 | 0.009486 |
| Val | 0.256764 | 0.261324 | 0.014816 | 0.017327 | 0.012412 | 0.216424 | 0.226765 | 0.011868 | 0.013984 | 0.010059 |
| Test | 0.243788 | 0.247817 | 0.014067 | 0.017386 | 0.011470 | 0.206514 | 0.215189 | 0.011513 | 0.014250 | 0.009212 |

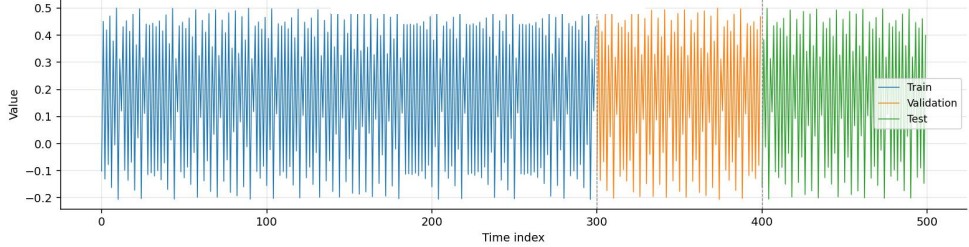

Figure 17: Gauss map time series with indications (vertical dashed lines) of the chronological train/validation/test split.

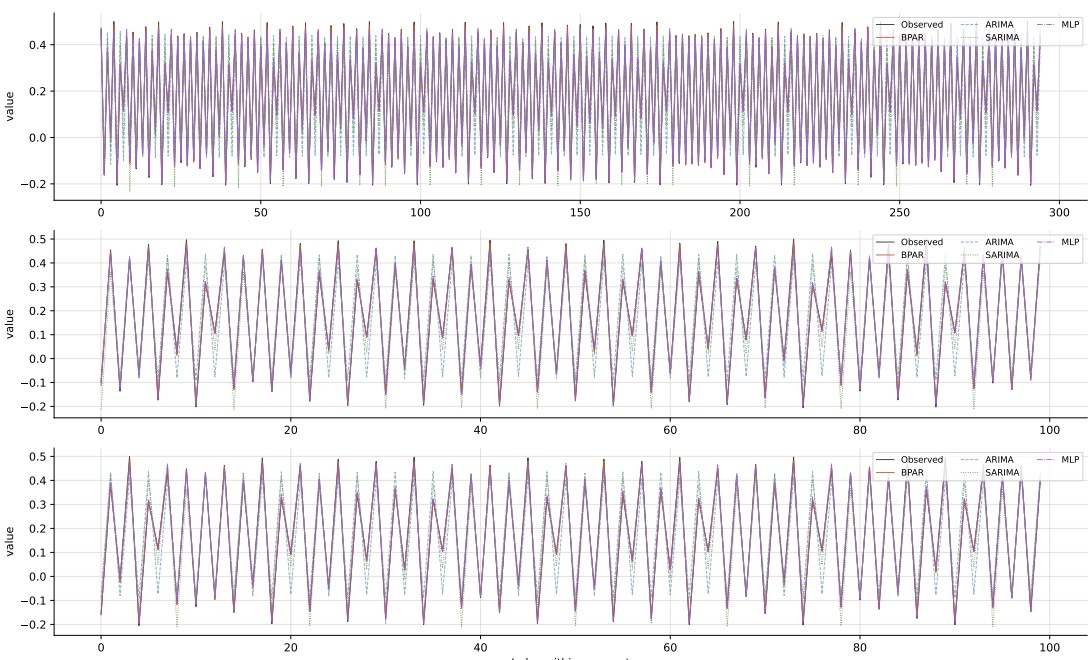

Figure 18: Train, validation, and test set predictions comparison for the Gauss map time series.

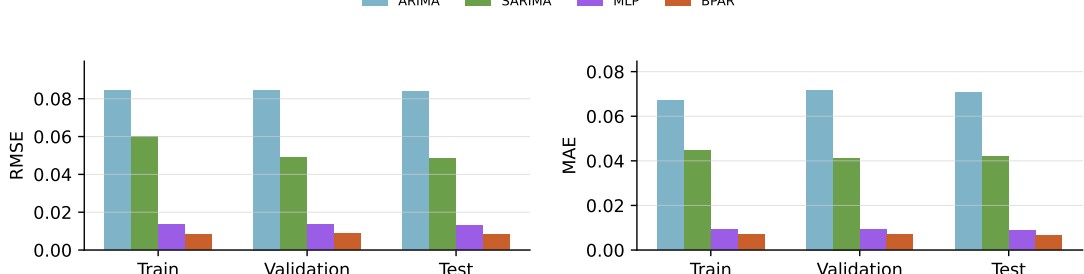

Figure 19: RMSE and MAE comparison across the train, validation, and test splits for the Gauss map benchmark.

Table 5: RMSE and MAE comparison for the Gauss map time series across train, validation, and test splits.

| Split | ARIMA RMSE | SARIMA RMSE | MLP RMSE | BPAR RMSE | ARIMA MAE | SARIMA MAE | MLP MAE | BPAR MAE |
|---|---|---|---|---|---|---|---|---|
| Train | 0.084599 | 0.060093 | 0.013300 | 0.008423 | 0.066850 | 0.044778 | 0.009104 | 0.006914 |
| Val | 0.084605 | 0.049018 | 0.013261 | 0.008704 | 0.071754 | 0.041287 | 0.009150 | 0.007036 |
| Test | 0.084016 | 0.048566 | 0.012987 | 0.008275 | 0.070664 | 0.042094 | 0.008936 | 0.006567 |

### A.2.3 Mackey–Glass delay system

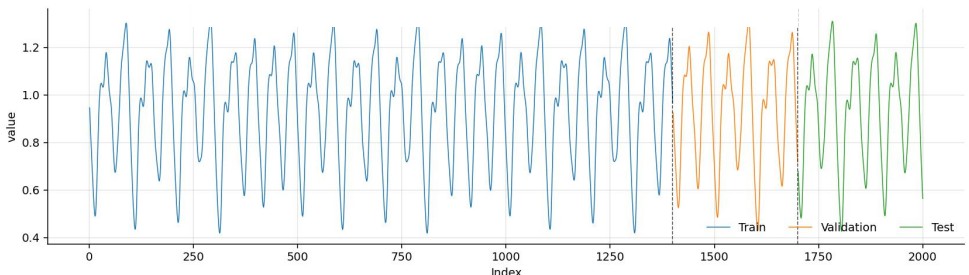

Figure 20: Mackey–Glass time series with indications (vertical dashed lines) of the chronological train/validation/test split.

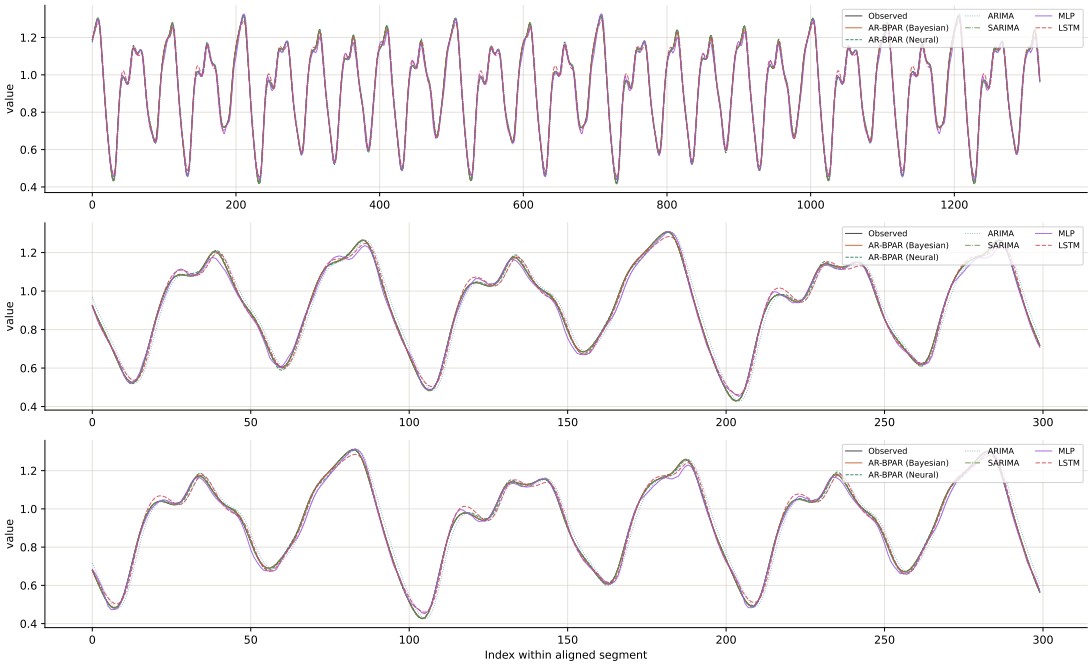

Figure 21: Train, validation, and test set predictions comparison for the Mackey-Glass time series.

Table 6: RMSE and MAE comparison for the Mackey-Glass time series across train, validation, and test splits.

| Split | Bayes AR-BPAR RMSE | Neural AR-BPAR RMSE | ARIMA RMSE | SARIMA RMSE | MLP RMSE | LSTM RMSE | Bayes AR-BPAR MAE | Neural AR-BPAR MAE | ARIMA MAE | SARIMA MAE | MLP MAE | LSTM MAE |
|---|---|---|---|---|---|---|---|---|---|---|---|---|
| Train | 0.00173 | 0.00654 | 0.04157 | 0.04950 | 0.01894 | 0.01868 | 0.00130 | 0.00478 | 0.02818 | 0.00557 | 0.01472 | 0.01570 |
| Validation | 0.00164 | 0.00611 | 0.03314 | 0.00336 | 0.01808 | 0.01794 | 0.00126 | 0.00459 | 0.02817 | 0.00273 | 0.01438 | 0.01517 |
| Test | 0.00162 | 0.00619 | 0.03297 | 0.00343 | 0.01790 | 0.01811 | 0.00123 | 0.00455 | 0.02781 | 0.00280 | 0.01422 | 0.01528 |

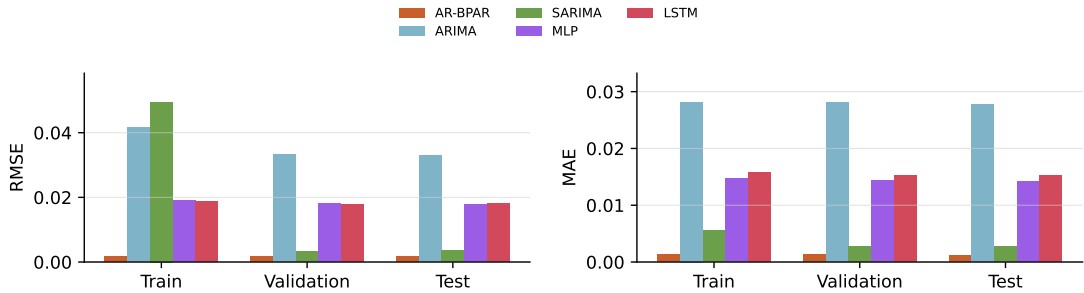

Figure 22: RMSE and MAE comparison across the train, validation, and test splits for the Mackey–Glass benchmark.

### A.3    Real data time series

Table 7: Held-out test set prediction comparison for the real data time series. LB refers to the residual Ljung–Box test $p$-value.

| Dataset | Method | Test RMSE | Test MAE | Test LB p |
|---|---|---|---|---|
| Ontario electricity | AR-BPAR (Bayesian) | 9.78 | 6.27 | 0.39 |
| Ontario electricity | AR-BPAR (Neural) | 9.57 | 5.96 | 0.77 |
| Ontario electricity | ARIMA | 9.58 | 5.51 | 0.79 |
| Ontario electricity | SARIMA | 9.77 | 6.11 | 0.23 |
| Ontario electricity | MLP | 10.81 | 7.19 | 0.98 |
| Ontario electricity | LSTM | 9.47 | 6.22 | 0.92 |
| Beijing PM$_{2.5}$ | ARIMA | 37.00 | 19.13 | 0.26 |
| Beijing PM$_{2.5}$ | SARIMA | 37.01 | 19.35 | 0.31 |
| Beijing PM$_{2.5}$ | MLP | 36.41 | 18.97 | 0.34 |
| Beijing PM$_{2.5}$ | LSTM | 36.19 | 18.59 | 0.61 |
| Beijing PM$_{2.5}$ | AR-BPAR (Bayesian) | 36.13 | 18.69 | 0.59 |

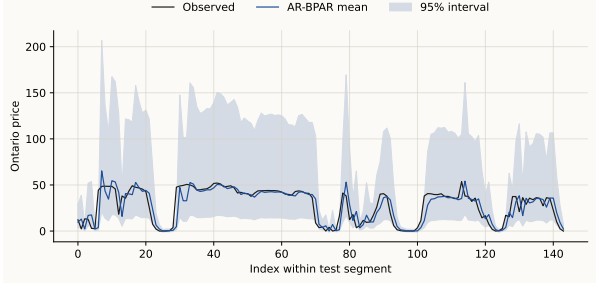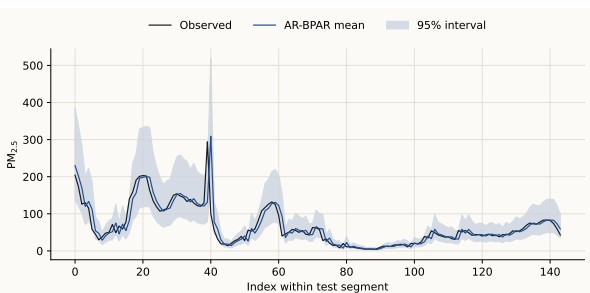

Figure 23: Held-out test set predictions along with their posterior prediction intervals. The left panel is Ontario electricity prices, and the right panel is Beijing PM$_{2.5}$.

### A.3.1    Ontario day-ahead electricity prices

Table 8: Prediction comparison for the Ontario electricity price time series across train, validation, and test splits.

| Split | Bayes AR-BPAR RMSE | Bayes AR-BPAR MAE | Neural AR-BPAR RMSE | Neural AR-BPAR MAE | ARIMA RMSE | ARIMA MAE | SARIMA RMSE | SARIMA MAE | MLP RMSE | MLP MAE | LSTM RMSE | LSTM MAE |
|---|---|---|---|---|---|---|---|---|---|---|---|---|
| Train | 38.76 | 10.16 | 39.91 | 9.85 | 38.98 | 9.73 | 42.36 | 10.38 | 37.38 | 9.88 | 39.45 | 10.14 |
| Validation | 14.48 | 8.74 | 14.15 | 8.51 | 14.47 | 8.61 | 14.40 | 8.78 | 14.00 | 9.02 | 13.86 | 8.56 |
| Test | 9.78 | 6.27 | 9.57 | 5.96 | 9.58 | 5.51 | 9.77 | 6.11 | 10.81 | 7.19 | 9.47 | 6.22 |

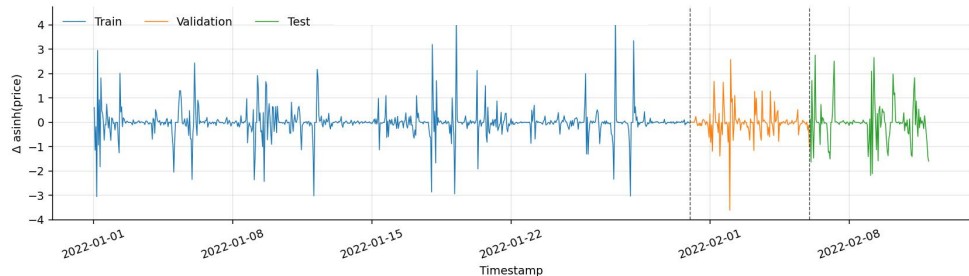

Figure 24: Ontario electricity price time series with indications (vertical dashed lines) of train/validation/test splits.

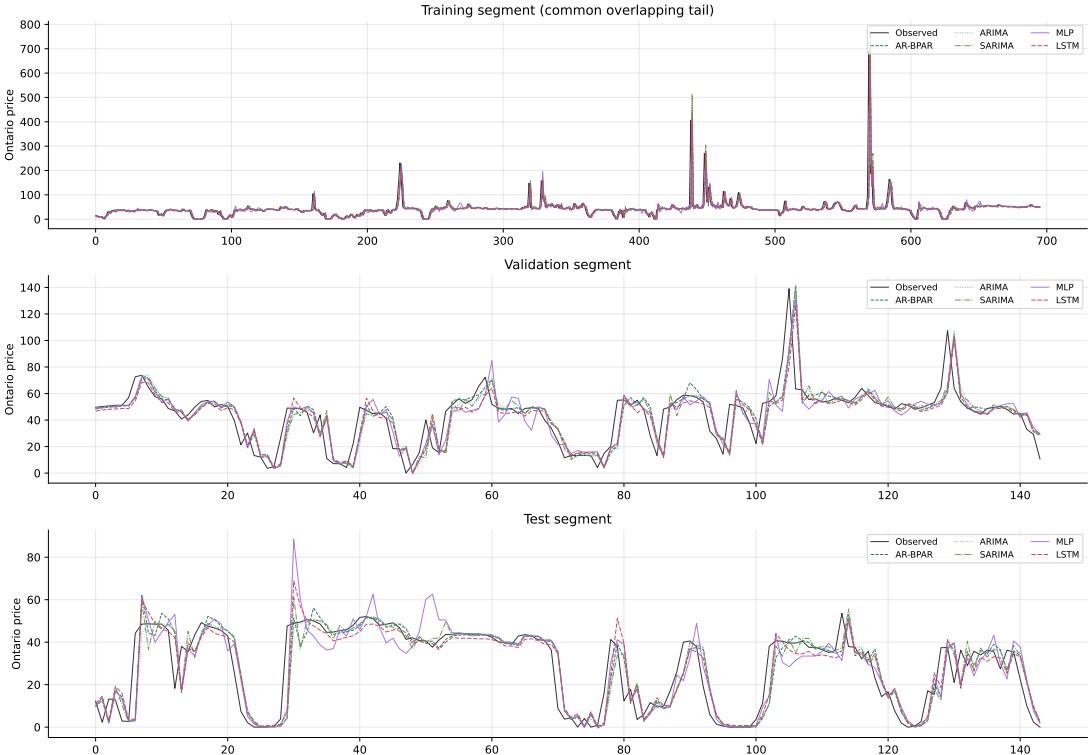

Figure 25: Train, validation, and test set predictions comparison for the Ontario electricity price time series.

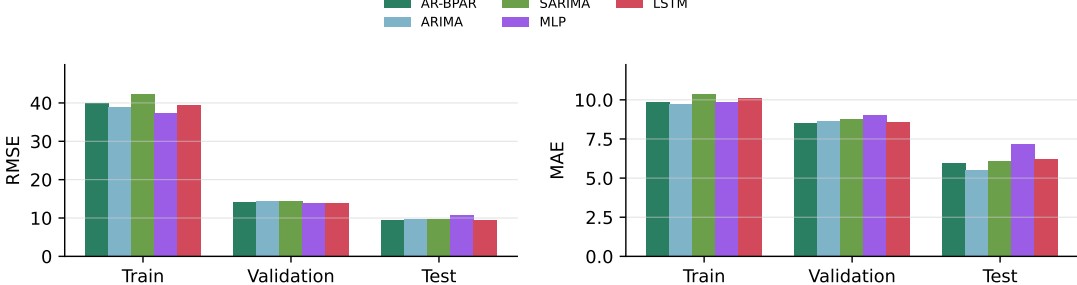

Figure 26: RMSE and MAE comparison across the train, validation, and test splits for the Ontario electricity price time series.

## A.3.2  Beijing PM$_{2.5}$

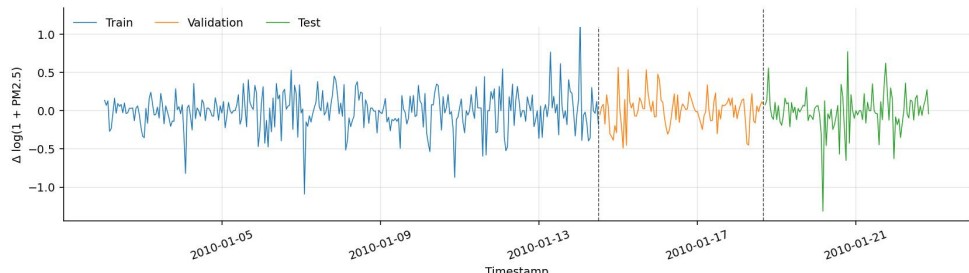

Figure 27: PM$_{2.5}$ time series with indications (vertical dashed lines) of train/validation/test splits.

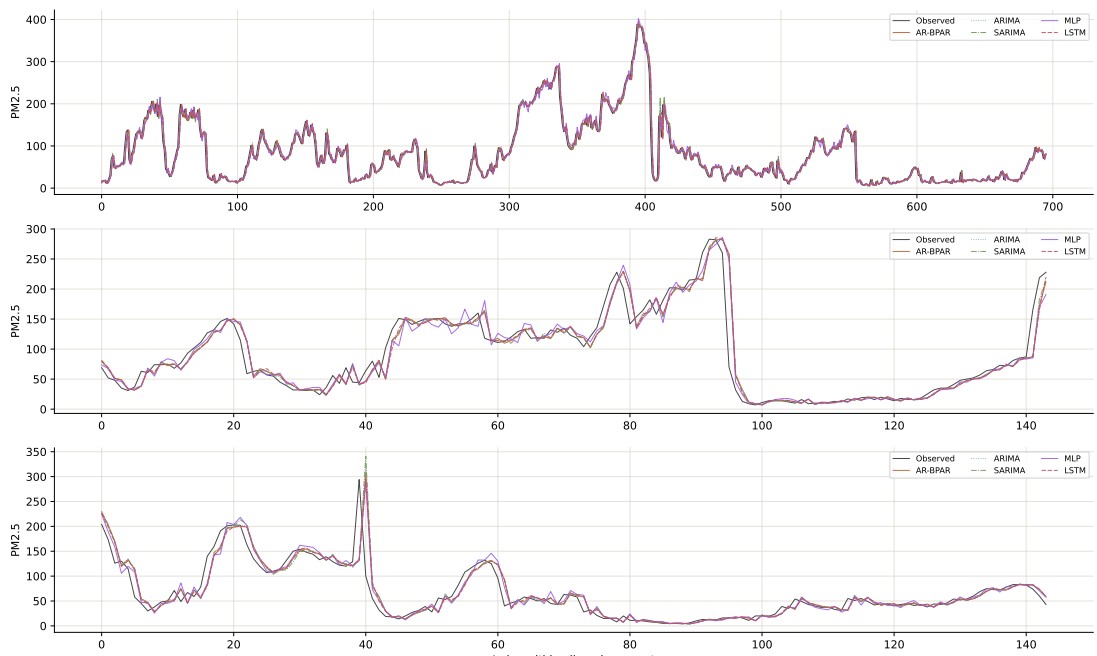

Figure 28: Train, validation, and test set predictions comparison for the PM$_{2.5}$ time series.

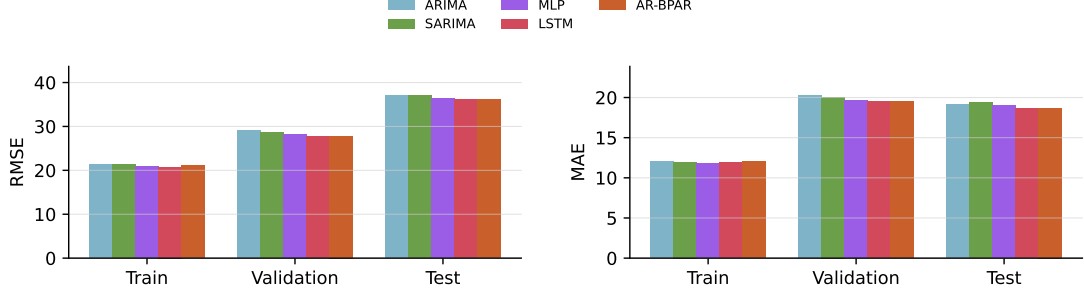

Figure 29: RMSE and MAE comparison across the train, validation, and test splits for the log-differenced PM$_{2.5}$ time series.

Table 9: Prediction comparison for the Ontario electricity price time series across train, validation, and test splits.

| Split | ARIMA RMSE | SARIMA RMSE | MLP RMSE | LSTM RMSE | AR-BPAR RMSE | ARIMA MAE | SARIMA MAE | MLP MAE | LSTM MAE | AR-BPAR MAE |
|---|---|---|---|---|---|---|---|---|---|---|
| Train | 21.40 | 21.36 | 20.80 | 20.79 | 21.06 | 12.02 | 11.98 | 11.77 | 11.90 | 12.02 |
| Validation | 29.03 | 28.66 | 28.23 | 27.72 | 27.85 | 20.23 | 20.05 | 19.59 | 19.46 | 19.46 |
| Test | 37.00 | 37.01 | 36.41 | 36.19 | 36.13 | 19.13 | 19.35 | 18.97 | 18.59 | 18.69 |

### A.4 Selected model configurations

For the nineteen (19) real-world datasets considered, this section provides the selected model configuration. Note that for each dataset, we apply the technique written in Section 2.4 to select the optimal BPAR or AR-BPAR model.

Further, we also report the effective degrees of freedom which serves as a proxy for the chosen model configuration's size. The matrix $S$ is written as

$$S = H\left(H^\top H + V_0^{-1}\right)^{-1} H^\top, \tag{28}$$

where $H$ is the design matrix and $V_0$ is the prior covariance matrix; see Section 2.3.2. Following (Hastie & Tibshirani, 1990), the effective degrees of freedom is defined as

$$\text{EDoF} = \text{tr}(S) = \text{tr}\left(H\left(H^\top H + V_0^{-1}\right)^{-1} H^\top\right). \tag{29}$$

Observe that with no shrinkage ($V_0^{-1} \to 0$) this measure equals the number of columns of $H$. As the shrinkage prior tightens, the EDoF decreases toward zero. It therefore measures the number of parameters the prior effectively lets the model spend. Table 10 also reports the selected Bernstein polynomial degree across the 19 real-world datasets. We witness this polynomial degree range from $m = 5$ to $m = 40$.

Table 10: Selected AR-BPAR model configuration based on the procedure described in Section 2.4. Note that $h$ is input scaling map (min–max or logistic, cf. Section 2.2). $\tau_B, \gamma_m$ refer to the selected prior parameters; EDoF denotes effective degrees of freedom (Hastie & Tibshirani, 1990).

| Dataset | $d_L$ | $d_B$ | $m$ | $h$ | $\tau_B$ | $\gamma_m$ | EDoF |
|---|---|---|---|---|---|---|---|
| Appliances energy | 6 | 3 | 20 | logistic | 5 | 0.5 | 31.8 |
| Australian demand | 24 | 6 | 40 | min–max | 5 | 0 | 95.4 |
| Beijing multi-site | 1 | 1 | 5 | min–max | 5 | 0 | 5.0 |
| Beijing PM2.5 | 12 | 6 | 5 | logistic | 0.5 | 0 | 24.1 |
| Blowflies | 8 | 4 | 5 | logistic | 5 | 0.5 | 16.3 |
| Canadian lynx | 1 | 2 | 8 | logistic | 2 | 0 | 6.6 |
| Colorado streamflow | 24 | 6 | 40 | logistic | 2 | 0.5 | 59.9 |
| Electricity load | 1 | 6 | 40 | logistic | 5 | 0 | 73.7 |
| Heart rate | 24 | 1 | 10 | logistic | 5 | 0 | 30.1 |
| Household power | 1 | 3 | 40 | logistic | 5 | 0 | 32.7 |
| Lorenz $x$ | 8 | 6 | 40 | min–max | 5 | 0 | 62.7 |
| Mackey–Glass | 8 | 6 | 20 | logistic | 5 | 0 | 32.9 |
| Melbourne temp. | 12 | 6 | 3 | logistic | 2 | 0 | 24.8 |
| Ontario electricity | 1 | 3 | 20 | min–max | 2 | 0 | 18.8 |
| Rössler $x$ | 2 | 6 | 10 | min–max | 5 | 0.5 | 33.6 |
| Santa Fe laser | 12 | 7 | 40 | min–max | 2 | 0.5 | 59.6 |
| Solar power | 3 | 3 | 10 | logistic | 5 | 0.5 | 22.6 |
| Sunspots (monthly) | 48 | 3 | 5 | logistic | 0.5 | 0.5 | 53.5 |
| Traffic (hourly) | 24 | 3 | 40 | logistic | 5 | 0 | 55.6 |

Finally, we report the criteria used to select the configurations of the competing MLP and LSTM methods; every method is tuned under the same chronological train/validation/test protocol. For the MLP, we perform a grid search over the input lag order in $\{3, 6, 12, 24\}$, the hidden-layer architecture in $\{(16), (32), (32, 16)\}$, and the $\ell_2$ penalty in $\{10^{-4}, 10^{-3}\}$. Each candidate is a feedforward network with ReLU activations trained on standardized inputs by the Adam optimizer (Kingma & Ba, 2015) with initial learning rate $10^{-3}$, for at most 2,000 iterations with validation-based early stopping. For the LSTM, we perform a grid search

over the input sequence length in $\{12, 24, 48\}$, the hidden state size in $\{8, 16\}$, and the learning rate in $\{10^{-3}, 3 \times 10^{-4}\}$. Each candidate is a single-layer LSTM with a linear output head trained on standardized inputs by Adam (weight decay $10^{-4}$) for up to 250 epochs, with early stopping on the validation loss (patience 30) and gradient-norm clipping. In both cases, the configuration attaining the lowest validation RMSE is selected. The capacity-matched variants, denoted "(matched)" in the tables of Appendix B, repeat the identical grid search with the hidden sizes enlarged so that the trainable parameter count matches or exceeds that of the selected AR-BPAR model on the same dataset. The linear and classical baselines follow the same rule: the ridge AR model is selected over lag orders $\{1, 3, 6, 12, 24\}$ and ridge penalties $\{0.1, 1, 10\}$, and the ARIMA/SARIMA models over the orders $p \in \{0, 1, 2, 3\}$, $d \in \{0, 1\}$, $q \in \{0, 1, 2\}$ (plus seasonal orders $(P, D, Q) \in \{0, 1\}^3$ at the dataset's seasonal period for SARIMA), each estimated by maximum likelihood in `statsmodels` and selected by the lowest validation RMSE.

### A.5 Detecting linear/nonlinear temporal dependence in practice

In this section, we build on the posterior credible intervals on the Bernstein weight coefficients provided by the method from Section 2.3.2. From Table 10, the Ontario electricity price time series has selected model orders $d_L = 1$, $d_B = 3$ and Bernstein polynomial degree $m = 20$. For $j = 1, 2, 3$, the Bernstein weight coefficients $w_{j,k}$ has a 95% posterior credible interval and this can then be used to generate a credible interval for $f_j(z) = \sum_{k=1}^m w_{j,k} B_{k,m}(z)$. This $f_j(z)$ corresponds to the fitted time lag level response. With $d_B = 3$, Figure 30 plots the three fitted lag responses $f_j(z)$ along with their posterior 95% credible intervals, where $z = h(x) \in [0, 1]$ is the min–max scaled value of the observed time series. We notice that at time lag 2, the credible interval is significantly away from zero. With this time series, the posterior credible intervals also revealed insignificance in the linear weight coefficient corresponding to $d_L = 1$. Next, with selected order $d_B = 6$ for the Beijing PM2.5 time series, Figure 31 shows the corresponding plot for that time series. The Ontario electricity price time series showed significant nonlinearity in temporal dependence, especially at time lag 2, whereas the Beijing PM2.5 does not reveal any significant nonlinearity.

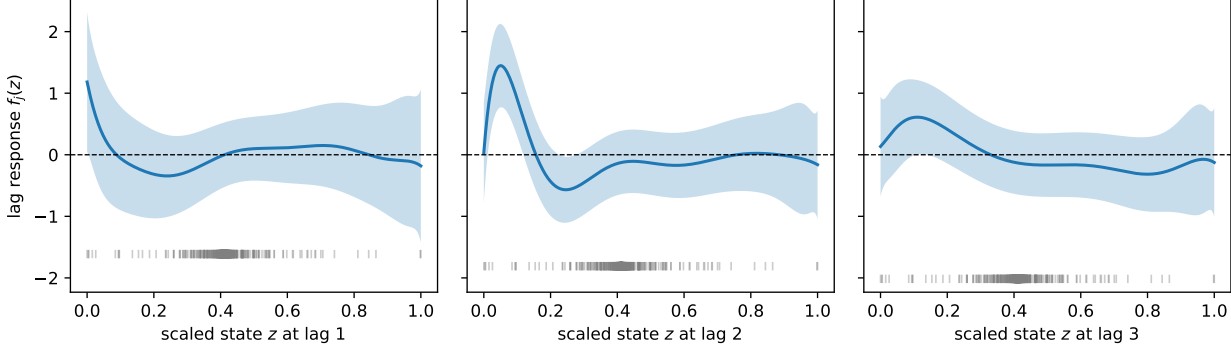

Figure 30: Fitted time lag responses $f_j(z) = \sum_{k=1}^m w_{j,k} B_{k,m}(z)$, $j = 1, 2, 3$, of the selected AR-BPAR model on the Ontario electricity price time series. Plot includes 95% posterior credible bands (shaded).

## B Additional empirical study results with real-world datasets

Table 11 reports, per dataset, held-out test RMSE relative to the best method on that dataset. Table 12 shows the resulting overall ranking of different methods in terms of forecasting performance. In these tables, "MLP (matched)" and "LSTM (matched)" denote the capacity-matched variants: the same MLP/LSTM architectures re-tuned with hidden sizes enlarged so that their trainable-parameter count matches or exceeds the selected BPAR/AR-BPAR model's model configuration; see also Appendix B.2.

We see that across the 19 datasets, winners are spread over different methods. However, it must be noted that the new BPAR/AR-BPAR is consistently best-or-near-best in most cases. Table 12 reports the mean rank of the various methods along with the median relative test RMSE. Overall, we witness the strong performance of the AR-BPAR method. Note that in Table 12 and elsewhere in the Appendix, denoting

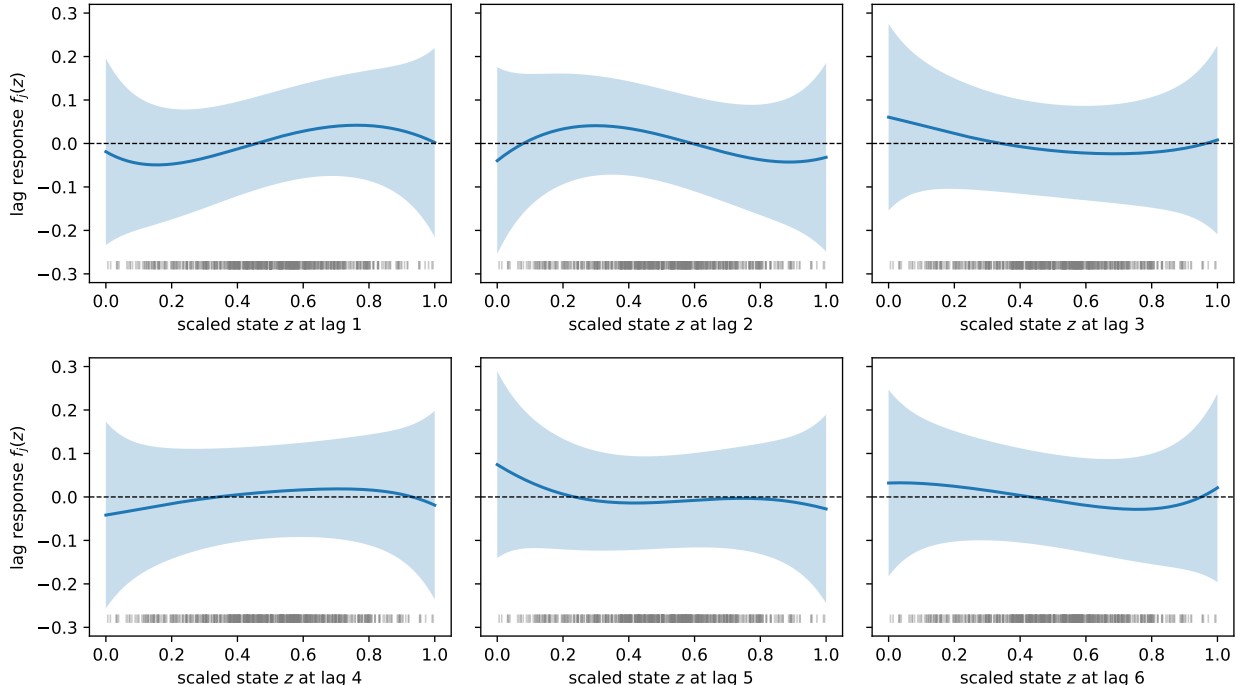

Figure 31: Fitted time lag responses $f_j(z) = \sum_{k=1}^{m} w_{j,k} B_{k,m}(z)$, $j = 1, 2, 3$, of the selected AR-BPAR model on the Beijing PM2.5 time series. Plot includes 95% posterior credible bands (shaded).

BPAR/AR-BPAR with "(nn)" refers to our models being estimated via the backpropagation technique from Section 2.3.1.

Table 11: Test RMSE based performance relative to best method. 1.00 = best, denoted in bold.

| Dataset | AR-BPAR | BPAR | RidgeAR | ARIMA | MLP | MLP (matched) | LSTM | N-BEATS | DeepAR | Transformer | N-HiTS |
|---|---|---|---|---|---|---|---|---|---|---|---|
| Blowflies | **1.00** | 1.14 | 1.00 | 2.49 | 4.55 | 4.55 | 1.46 | 1.31 | 2.55 | 1.34 | 1.04 |
| Canadian lynx | 1.00 | **1.00** | 1.17 | 2.68 | 1.20 | 1.21 | 1.31 | 1.10 | 2.70 | 1.18 | 1.22 |
| Colorado streamflow | **1.00** | 1.46 | 1.06 | 2.65 | 2.03 | 1.68 | 2.04 | 1.13 | 1.47 | 1.45 | 1.10 |
| Household power | 1.00 | 1.00 | 1.39 | 1.24 | 1.41 | 1.34 | 1.05 | 1.07 | 1.11 | 1.16 | 1.14 |
| Beijing multi-site | 1.01 | 1.01 | **1.00** | 1.01 | 1.02 | 1.02 | 1.02 | 1.18 | 1.04 | 1.03 | 1.02 |
| Melbourne temp. | 1.02 | 1.02 | 1.00 | 2.57 | 1.04 | **1.00** | 1.06 | 1.13 | 1.00 | 1.00 | 1.17 |
| Beijing PM2.5 | 1.03 | 1.02 | **1.00** | 1.03 | 1.22 | 1.01 | 1.03 | 1.18 | 1.00 | 1.00 | 1.16 |
| Sunspots (monthly) | 1.03 | 1.09 | 1.02 | 4.65 | 1.03 | 1.05 | 1.11 | 1.20 | **1.00** | 1.06 | 1.15 |
| Appliances energy | 1.04 | 1.05 | 1.01 | 1.06 | 1.03 | 1.05 | 1.05 | 1.14 | **1.00** | 1.01 | 1.05 |
| Electricity load | 1.05 | 1.05 | 1.00 | 1.07 | 1.07 | 1.04 | 1.02 | 1.30 | 1.00 | **1.00** | 1.25 |
| Ontario electricity | 1.06 | 1.06 | 1.03 | 1.30 | 1.13 | 1.14 | 1.11 | 1.31 | 1.02 | 1.04 | 1.15 |
| Heart rate | 1.10 | 1.08 | **1.00** | 4.19 | 1.31 | 1.52 | 1.44 | 1.06 | 1.02 | 1.13 | 1.04 |
| Lorenz $x$ | 1.08 | 1.38 | 1.31 | 83.47 | 2.04 | 1.83 | 4.65 | 1.43 | **1.00** | 2.85 | 1.35 |
| Solar power | 1.16 | 1.17 | 1.55 | 2.12 | 1.34 | 1.66 | 1.17 | **1.00** | 1.29 | 1.02 | 1.06 |
| Rössler $x$ | 1.24 | 1.25 | 2.02 | 34.90 | 2.00 | 2.26 | 2.41 | 1.06 | **1.00** | 2.48 | 1.23 |
| Traffic (hourly) | 1.42 | 1.47 | 1.75 | 2.54 | 1.61 | 1.22 | 1.20 | 1.04 | 2.35 | **1.00** | 1.33 |
| Mackey–Glass | 1.79 | 1.88 | 2.19 | 40.11 | 2.67 | 2.61 | 4.12 | **1.00** | 1.34 | 2.75 | 1.19 |
| Australian demand | 1.97 | 2.04 | 2.30 | 5.08 | 2.39 | 2.00 | 2.45 | 1.03 | 1.89 | 1.59 | **1.00** |
| Santa Fe laser | 2.68 | 3.53 | 7.12 | 10.90 | 2.08 | 2.80 | 5.57 | 1.04 | 5.89 | 2.60 | **1.00** |

## B.1 Measuring quality of posterior prediction intervals

In this section, we discuss certain measures used for assessing the quality of prediction intervals obtained by applying the Bayesian procedure from Section 2.3. First, we consider Prediction Interval Coverage Probability (PICP) that measures the fraction of test points that the interval captures. A well-calibrated model has $\text{PICP}(\alpha) \approx 1 - \alpha$; see Khosravi et al. (2011) for details. Second, the mean relative prediction interval width ($\text{MPIW}_{\text{rel}}$) from Khosravi et al. (2011) which speaks to the sharpness of the interval. PICP and $\text{MPIW}_{\text{rel}}$ must be read together: coverage is only meaningful at a stated sharpness, since an arbitrarily wide interval attains $\text{PICP} = 1$ trivially. Third, the continuous ranked probability score (CRPS) that assesses the entire predictive distribution against the realized value; see Gneiting & Raftery (2007) for details.

Table 12: Overall accuracy ranking and median relative RMSE to the per-dataset best performing method. Results based on 19 real-world datasets.

| Method | Datasets | Mean rank | Median relative RMSE |
|---|---|---|---|
| **AR-BPAR** | 19 | 4.89 | 1.052 |
| RidgeAR | 19 | 5.58 | 1.057 |
| Transformer | 19 | 5.84 | 1.133 |
| DeepAR | 19 | 6.11 | 1.039 |
| **BPAR** | 19 | 6.16 | 1.094 |
| N-HiTS | 19 | 7.00 | 1.146 |
| N-BEATS | 19 | 7.47 | 1.130 |
| AR-BPAR (nn) | 19 | 7.53 | 1.148 |
| MLP (matched) | 19 | 8.42 | 1.337 |
| LSTM | 19 | 9.26 | 1.195 |
| MLP | 19 | 10.00 | 1.339 |
| BPAR (nn) | 19 | 10.05 | 1.472 |
| LSTM (matched) | 19 | 11.74 | 1.898 |
| SARIMA | 18 | 12.92 | 2.609 |
| Naive | 19 | 13.11 | 1.610 |
| ETS | 19 | 13.32 | 2.850 |
| ARIMA | 19 | 13.97 | 2.568 |
| SeasonalNaive | 8 | 16.50 | 1.899 |

Table 13 reports, per dataset, the PICP at the coverage 90% and 95% levels, the relative MPIW, and the CRPS. Note that for BPAR/AR-BPAR models, these metrics evaluate the exact Student-$t$ posterior predictive intervals; see Section 2.3.2. In this table, only DeepAR's CRPS is displayed alongside for comparison because DeepAR is a competing method with a native predictive distribution at comparable point accuracy, and also is the strongest probabilistic baseline in the study (best CRPS on 13 of 19 datasets). Hence, it provides the most demanding like-for-like reference for BPAR/AR-BPAR's interval quality. ARIMA/SARIMA analytic intervals are summarized in the table footer. DeepAR's intervals are computed as follows: the model is trained with a Gaussian likelihood on the network output, and its predictive distribution at each test step is obtained by ancestral sampling of full trajectories from the fitted model (Salinas et al., 2017), as implemented in the `darts` package (Herzen et al., 2022); we draw 500 sample paths per rolling one-step forecast, summarize them by their sample mean and standard deviation, form the 90%/95% intervals from the corresponding Gaussian quantiles, and evaluate the CRPS with the closed-form expression for a Gaussian predictive intervals (Gneiting & Raftery, 2007). We see that BPAR/AR-BPAR attains the best CRPS at 6 of 19 datasets, but delivers a performance that is highly competitive with DeepAR. Across the 19 real-world datasets, AR-BPAR's 90% intervals attain a median empirical coverage of 0.92. It must be emphasized that the new BPAR/AR-BPAR model operates at a small fraction of the parameter count used by DeepAR; see Table 14 for the exact counts.

As an illustration, we consider two highly nonlinear time series datasets from our list (Santa Fe laser and Rössler) and plot the nominal and empirical coverage of the posterior predictive intervals; see Figure 32.

### B.2 Parameter counts from competing methods

Table 14 reports, per dataset, the trainable-parameter count of the selected AR-BPAR model alongside the minimum and maximum trainable-parameter counts among the neural network/deep learning methods considered (MLP, LSTM, N-BEATS, DeepAR, Transformer, N-HiTS). This table also presents the head-to-head test RMSE of BPAR/AR-BPAR versus the best neural network/deep learning method.

Table 14 shows that even the smallest model configuration among the neural network/deep learning competitors is larger than our AR-BPAR's size, and this underscores the parsimony of the proposed model without

Table 13: Performance of posterior predictive intervals on held-out test sets. PICP@90/95 = empirical coverage of nominal 90%/95% intervals; $\text{MPIW}_{\text{rel}}$ = mean relative interval width; CRPS (lower is better; bold = better of BPAR vs. DeepAR).

| | AR-BPAR | | | | DeepAR |
|---|---|---|---|---|---|
| Dataset | PICP@90 | PICP@95 | $\text{MPIW}_{\text{rel}}$ | CRPS | CRPS |
| Appliances energy | 0.92 | 0.97 | 3.88 | 0.201 | **0.190** |
| Australian demand | 1.00 | 1.00 | 5.71 | 0.015 | **0.006** |
| Beijing multi-site | 0.88 | 0.88 | 2.52 | **0.238** | 0.241 |
| Beijing PM2.5 | 0.83 | 0.90 | 2.03 | 0.211 | **0.205** |
| Blowflies | 0.93 | 0.97 | 1.28 | **359.671** | 1047.454 |
| Canadian lynx | 0.96 | 0.96 | 1.52 | **0.247** | 0.686 |
| Colorado streamflow | 0.96 | 0.97 | 1.90 | **0.032** | 0.043 |
| Electricity load | 0.90 | 0.91 | 3.18 | 0.195 | **0.182** |
| Heart rate | 0.93 | 0.95 | 0.62 | 0.701 | **0.653** |
| Household power | 0.88 | 0.92 | 2.32 | **0.166** | 0.178 |
| Lorenz $x$ | 0.91 | 0.96 | 0.14 | 0.194 | **0.176** |
| Mackey–Glass | 1.00 | 1.00 | 0.72 | 0.013 | **0.005** |
| Melbourne temp. | 0.93 | 0.95 | 2.32 | 1.377 | **1.352** |
| Ontario electricity | 0.84 | 0.89 | 2.38 | 0.380 | **0.342** |
| Rössler $x$ | 0.92 | 0.95 | 0.12 | 0.136 | **0.110** |
| Santa Fe laser | 0.84 | 0.85 | 0.45 | **6.078** | 8.872 |
| Solar power | 0.99 | 0.99 | 3.06 | 0.063 | **0.046** |
| Sunspots (monthly) | 0.82 | 0.90 | 0.98 | 9.629 | **9.129** |
| Traffic (hourly) | 0.99 | 1.00 | 5.39 | 0.016 | **0.008** |
| Median | 0.92 | 0.95 | 2.03 | | |

ARIMA/SARIMA median $\text{MPIW}_{\text{rel}}$ for reference: 3.15.

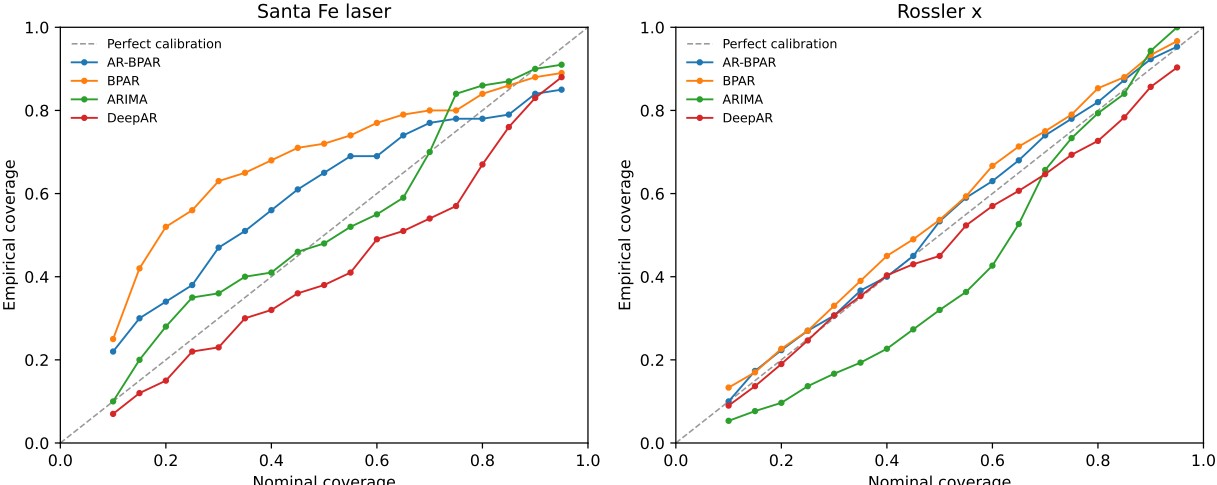

Figure 32: Reliability of posterior predictive intervals: nominal vs. empirical coverage. Diagonal means perfect calibration.

compromising too much on test RMSE. Figure 33 shows the resulting accuracy–size frontier: the proposed BPAR/AR-BPAR models reach top-tier accuracy at parameter counts that are orders of magnitude below the larger neural network/deep learning baselines.

Table 14: Model size and accuracy (test RMSE) across 19 real-world datasets. NN/DL denotes the set of neural network/deep learning competitors: MLP, LSTM, N-BEATS, DeepAR, Transformer, N-HiTS. Bold = better of AR-BPAR vs. the best neural baseline on test RMSE.

| | AR-BPAR | NN/DL | | Test RMSE | |
|---|---|---|---|---|---|
| Dataset | | Min | Max | AR-BPAR | Best NN/DL |
| Appliances energy | 70 | 161 | 845387 | 0.371 | **0.356** |
| Australian demand | 271 | 673 | 896659 | 0.012 | **0.006** |
| Beijing multi-site | 8 | 961 | 896659 | **0.479** | 0.483 |
| Beijing PM2.5 | 49 | 961 | 819751 | 0.432 | **0.421** |
| Blowflies | 33 | 161 | 810523 | **721.496** | 746.775 |
| Canadian lynx | 20 | 129 | 802319 | **0.436** | 0.480 |
| Colorado streamflow | 271 | 257 | 819751 | **0.058** | 0.064 |
| Electricity load | 248 | 97 | 896659 | 0.381 | **0.362** |
| Heart rate | 36 | 673 | 819751 | 1.716 | **1.601** |
| Household power | 125 | 257 | 819751 | **0.299** | 0.313 |
| Lorenz $x$ | 255 | 961 | 828979 | 0.385 | **0.358** |
| Mackey–Glass | 135 | 161 | 845387 | 0.011 | **0.006** |
| Melbourne temp. | 37 | 449 | 896659 | 2.460 | **2.424** |
| Ontario electricity | 65 | 257 | 819751 | 0.760 | **0.727** |
| Rössler $x$ | 69 | 673 | 845387 | 0.242 | **0.194** |
| Santa Fe laser | 300 | 961 | 819751 | 12.774 | **4.759** |
| Solar power | 37 | 673 | 819751 | 0.123 | **0.106** |
| Sunspots (monthly) | 67 | 1233 | 896659 | 17.422 | **16.852** |
| Traffic (hourly) | 148 | 673 | 819751 | 0.020 | **0.014** |

### B.3 Hyperparameter and prior distribution parameter sensitivity

Figures 34 and 35 concern hyperparameter sensitivity analysis on a representative dataset: the Santa Fe laser. First, a one-at-a-time sensitivity analysis of the hyperparameters $d_L$, $d_B$, and $m$ is considered. Here, per-axis refers to the one-at-a-time construction of these sweeps: each structural hyperparameter—the linear-lag order $d_L$, the Bernstein polynomial order $d_B$, and the polynomial degree $m$—defines one axis, and to trace it we vary that hyperparameter alone across its grid while holding the other two fixed at their optimally selected values based on the technique from Section 2.4. Note that each panel of Figure 34 is one such axis. The per-axis best validation RMSE is the lowest value attained anywhere along that single curve, which we compare against the value the selection rule (Section 2.4) actually chose on that axis. We emphasize that this is a local, one-at-a-time robustness check: it verifies that the selected configuration sits at or beside the best attainable value along each hyperparameter direction separately, and does not by itself certify global optimality of the joint triple $(d_L, d_B, m)$, since the axes are not swept in combination. Figure 34 reveals that reduction in RMSE is very minimal beyond a certain choice of the hyperparameters $d_L, d_B, m$. A similar story can be witnessed in the left plot in Figure 35 wherein we study sensitivity of RMSE as the prior distribution's parameter $\tau_B$ increases. Finally with the right plot in Figure 35, with $\tau_B$ fixed, we see that the effective degrees of freedom decreases as $\gamma$ increases. Recall that this $\gamma$ parameter directly controls the shrinkage of the Bernstein weight coefficients. Similarly, Figures 36 and 37 contain the corresponding sensitivity analysis plots for the Rössler time series dataset. The observations made with these two representative datasets were seen to broadly hold true for the other 17 real-world datasets considered in this article.

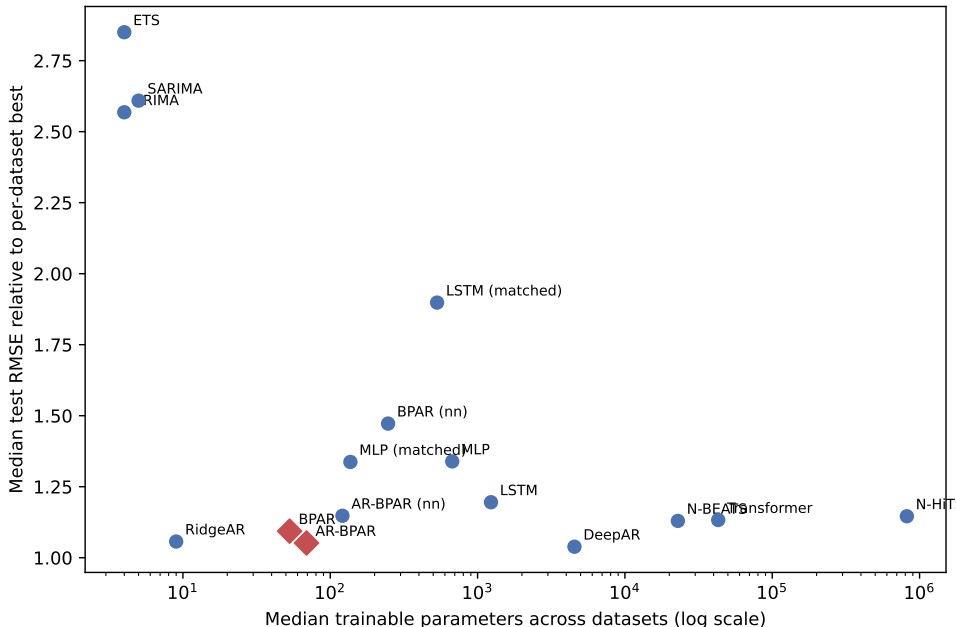

Figure 33: Accuracy–size frontier: median trainable parameters vs. median relative test RMSE across the 19 datasets.

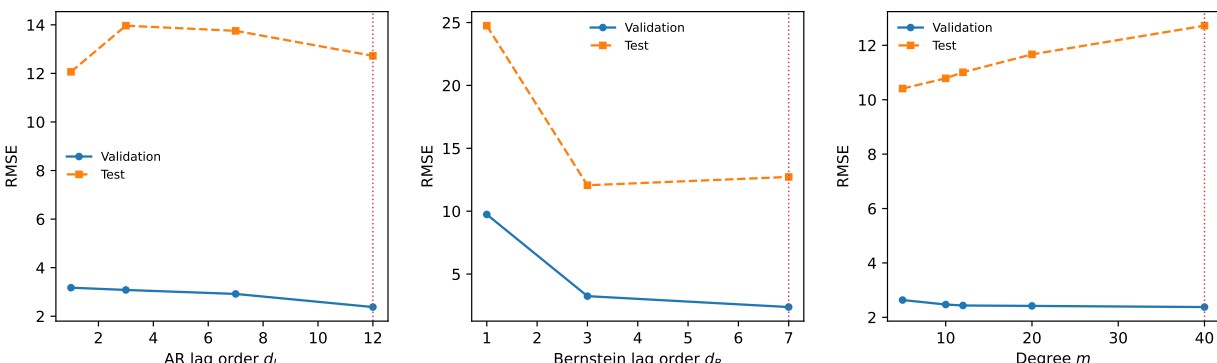

Figure 34: Hyperparameter sensitivity analysis on the Santa Fe laser dataset: validation/test RMSE as each structural hyperparameter is varied with the others held at their selected values.

### B.4 Model size and computing cost

With the exact Bayesian inference method in Section 2.3.2, the computation involves $K_\theta \times K_\theta$ sized matrix operations, and the associated cost grows polynomial with respect to the Bernstein polynomial degree $m$. Across the 19 real-world datasets considered in this article, the model size is seen to be $K_\theta \leq 300$. The exact Bayesian inference method from Section 2.3.2 takes at most 10 milliseconds. Further, the complete hyperparameter search involving the technique from Section 2.4 takes a median time of 9 seconds (maximum 31 seconds) per dataset on a single CPU core, with no GPU required. In comparison, the variational Bayesian inference route from Section 2.3.1 of the manuscript, estimated via backpropagation, is slower. A single model fit takes about 2 seconds on average, and the corresponding complete hyperparameter search takes a median time of approximately 2060 seconds (maximum approximately 4040 seconds) per dataset on the same single CPU core.

Table 14 shows that even the smallest model configuration among the neural network/deep learning competitors is larger than our AR-BPAR's model size, and this underscores the parsimony of the proposed model without compromising too much on prediction accuracy. Figure 33 shows the resulting accuracy–size

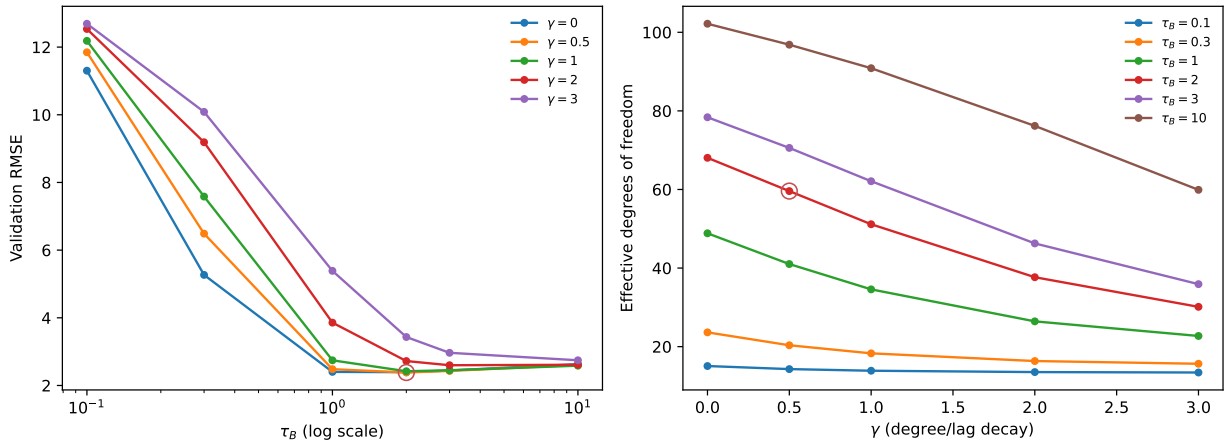

Figure 35: Prior distribution sensitivity on the representative Santa Fe laser dataset. Left: validation accuracy across $(\tau_B, \gamma)$. Right: effective degrees of freedom (Hastie & Tibshirani, 1990) fall monotonically in $\gamma$ at every $\tau_B$.

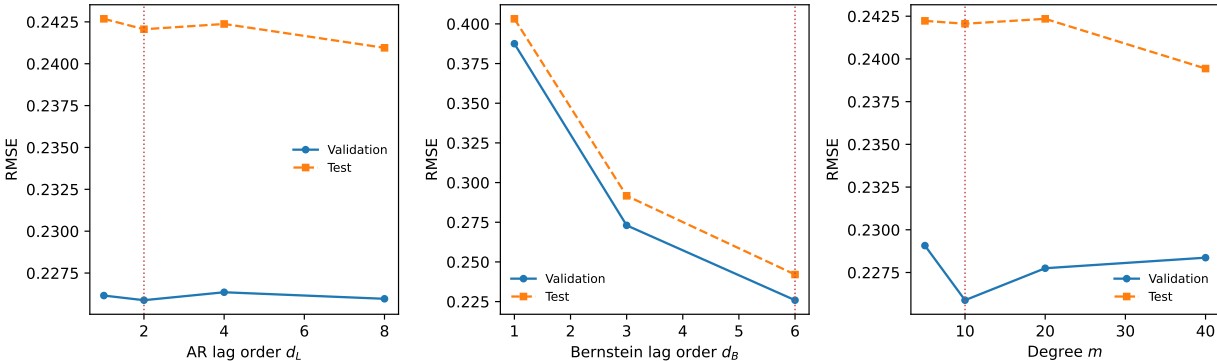

Figure 36: Hyperparameter sensitivity analysis on the Rössler dataset: validation/test RMSE as each structural hyperparameter is varied with the others held at their selected values.

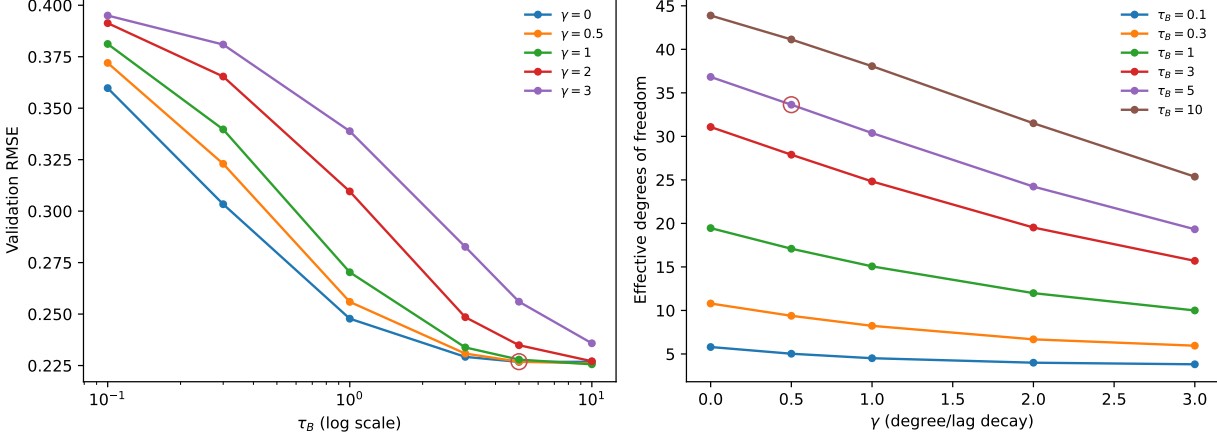

Figure 37: Prior distribution sensitivity on the Rössler dataset. Left: validation accuracy across $(\tau_B, \gamma)$. Right: effective degrees of freedom fall monotonically in $\gamma$ at every $\tau_B$.

frontier. There, we see that the proposed BPAR/AR-BPAR models reach top-tier accuracy at parameter counts that are orders of magnitude below the larger neural network/deep learning competitors.

### B.5 Real-world datasets and their sources

The expanded benchmark uses 19 datasets: 16 real-world measured series and three synthetic chaotic benchmarks. For each dataset the modeled variable name, the sampling span period, where we obtained the data, and the reference are listed as follows:

- **Ontario electricity prices** (hourly): 2022 day-ahead wholesale electricity prices (CAD/MWh) for the Ontario market, from the public data directory of the Independent Electricity System Operator (Independent Electricity System Operator, 2026); `https://www.ieso.ca/en/power-data/data-directory`.

- **Beijing PM$_{2.5}$** (hourly): PM$_{2.5}$ concentration measured at the U.S. Embassy in Beijing, 2010–2014, with meteorological covariates; UCI Machine Learning Repository (Dua & Graff, 2019; Chen, 2015); `https://archive.ics.uci.edu/dataset/381/beijing+pm2+5+data`.

- **Santa Fe laser (dataset A)**: far-infrared NH$_3$ laser intensity, dataset A of the Santa Fe Time Series Prediction Competition (Weigend & Gershenfeld, 1994); `https://www-psych.stanford.edu/~andreas/Time-Series/SantaFe.html`.

- **Monthly sunspots**: monthly mean international sunspot number, World Data Center SILSO, Royal Observatory of Belgium (SILSO World Data Center, 2024); `https://www.sidc.be/SILSO/datafiles`.

- **Heart rate** (0.5-second intervals): physiological heart-rate series distributed with the `darts` library (Herzen et al., 2022); `https://unit8co.github.io/darts/generated_api/darts.datasets.html`.

- **Individual household electric power** (one-minute): global active power of a single household in Sceaux, France, 2006–2010; UCI Machine Learning Repository (Dua & Graff, 2019); `https://archive.ics.uci.edu/dataset/235/individual+household+electric+power+consumption`.

- **Appliances energy** (ten-minute): appliances energy use of a low-energy house in Belgium (Candanedo et al., 2017); UCI Machine Learning Repository (Dua & Graff, 2019); `https://archive.ics.uci.edu/dataset/374/appliances+energy+prediction`.

- **Electricity load diagrams 2011–2014** (15-minute): load of a single Portuguese client from the ElectricityLoadDiagrams collection; UCI Machine Learning Repository (Dua & Graff, 2019); `https://archive.ics.uci.edu/dataset/321/electricityloaddiagrams20112014`.

- **Beijing multi-site air quality** (hourly): PM$_{2.5}$ at the Aotizhongxin monitoring station, 2013–2017 (Zhang et al., 2017); UCI Machine Learning Repository (Dua & Graff, 2019); `https://archive.ics.uci.edu/dataset/501/beijing+multi+site+air+quality+data`.

- **Solar power** (sub-hourly): solar power generation series from the Monash Time Series Forecasting Archive (Godahewa et al., 2021); `https://forecastingdata.org/`.

- **Australian electricity demand** (half-hourly): state-level electricity demand series from the Monash Time Series Forecasting Archive (Godahewa et al., 2021); `https://forecastingdata.org/`.

- **Traffic** (hourly): a single road-occupancy series from the San Francisco Bay Area traffic benchmark, via the Monash Time Series Forecasting Archive (Godahewa et al., 2021); `https://forecastingdata.org/`.

- **Nicholson's blowflies** (bi-daily): adult blowfly population counts from Nicholson's laboratory experiments (Nicholson, 1957), a classic nonlinear ecology benchmark; we use the digitization published in the ecological literature (`https://doi.org/10.1111/j.1442-9993.1977.tb01143.x`).

- **Canadian lynx** (annual, 1821–1934): number of lynx trapped in the Mackenzie River district (Elton & Nicholson, 1942), the classic threshold/nonlinear autoregression benchmark (Tong, 1990); obtained as `datasets::lynx` from R via the Rdatasets collection, `https://vincentarelbundock.github.io/Rdatasets/`.

- **Melbourne minimum temperature** (daily, 1981–1990): daily minimum temperatures in Melbourne recorded by the Australian Bureau of Meteorology; distributed as `daily-min-temperatures` at `https://github.com/jbrownlee/Datasets`.

- **Colorado River streamflow** (daily): mean daily discharge of the Colorado River at Lees Ferry, Arizona (USGS gauge 09380000), U.S. Geological Survey National Water Information System (U.S. Geological Survey, 2024); `https://waterservices.usgs.gov/`.

- **Lorenz, Rössler, and Mackey–Glass** (synthetic): the three chaotic benchmarks are generated by numerically integrating the standard equations—the Lorenz system with $(\sigma, \rho, \beta) = (10, 28, 8/3)$ (Lorenz, 1963), the Rössler system with $(a, b, c) = (0.2, 0.2, 5.7)$ (Rössler, 1976), and the Mackey–Glass delay equation with $\tau = 17$ (Mackey & Glass, 1977)—and adding observation noise with standard deviation equal to 2% of each series' standard deviation; the generation script is included with our code.

