# OpenReview forum: "Nonlinear Time Series Modeling Using Bernstein Polynomials and Bayesian Inference"
_TMLR — Under review for TMLR_

### Review · Reviewer_YeML · 2026-06-16

**Summary Of Contributions:**

This paper proposes a new method for forecasting nonlinear time series. Specifically, the paper uses Bernstein polynomials to model nonlinear autoregressions. The paper begins by pointing out the problems with existing methods: linear dependence in traditional models and lower interpretability and tuning cost in artificial neural networks. By adapting Bernstein polynomials to traditional modeling, the proposed method can overcome the linear dependence problem while providing higher interpretability of the forecast results. The paper mainly describes how the two methods (BPAR and AR-BPAR) work and how inference can be performed for them. Experiments on both artificial and real data demonstrate the proposed model’s performance.

### Strengths
1. The paper properly points out the problems with existing methods and is well-motivated.
2. The proposed method is simple and effectively improves the traditional forecasting models.
3. The proposed method provides closed-form outputs, enabling formal analysis of the results.
4. The writing is clear and easy to read, and contains a detailed explanation about the background.

### Weaknesses
1. The paper proposes a new approach. While the method is based on a well-known mathematical concept, its performance should be investigated more thoroughly, particularly using additional real-world datasets and more comparisons with modern forecasting methods.
    * Only two real-time series datasets are considered in the experiments. This is not enough to show the effectiveness of the proposed method in more general contexts.
    * Comparing the method with traditional models such as ARIMA and SARIMA is necessary, but the improvement is obvious because those models lack the ability to handle highly nonlinear data.
    * Comparing to MLP and LSTM would give us more insights (by comparing to neural network-based models). However, neither reflects the cutting-edge methods mentioned in the paper's introduction.
2. While the paper describes how to search for the best hyperparameters, the empirical study section does not provide the resulting hyperparameter choices. Also, more discussion on the hyperparameters would be helpful to general readers.

**Audience:**

Yes

**Audience Explanation:**

Time series forecasting is an important problem, and many industry practitioners are interested in new forecasting methods. As noted in the paper’s introduction, recent neural network-based models lack interpretability and require sampling to estimate confidence intervals. The proposed method provides clear closed-form outputs that can be mathematically analyzed to enable further post-processing. For these reasons, the proposed method has sufficient potential to interest TMLR’s audience.

**Broader Impact Concerns:**

I don’t see a particular broader impact concern regarding this paper.

**Claims And Evidence:**

No

**Claims Explanation:**

As noted in the Weaknesses section, the paper only compares with traditional methods (AR, ARIMA, and SARIMA) or simple neural networks (LSTM and MLP), and the experiments are mainly conducted on artificial data. Only two real-world datasets are used in the experiments, so the paper does not provide sufficient evidence that the proposed method generalizes to many real-world scenarios.

**Requested Changes:**

1. As noted in the Weaknesses section, this paper lacks sufficient experimental support to demonstrate its overall performance. Please add more experiments on more real-world datasets, with comparisons to more cutting-edge forecasting methods.
2. I have a few questions about the hyperparameter choices. Please consider adding some discussion about the answers to the questions.
    * In general, how large a Bernstein polynomial degree is sufficient?
    * What are the effects of different hyperparameter choices? For example, if a few hyperparameters are not chosen optimally, what would be the expected results?
    * Is there any trade-off for different hyperparameter choices? For example, does a large Bernstein polynomial degree imply higher computation cost to evaluate? Is there any trade-off between accuracy and efficiency?

---

> ### Author Response · Authors · 2026-07-15
> **Requested change 1: More real-world datasets and cutting-edge baselines**
>
> *As noted in the Weaknesses section, this paper lacks sufficient experimental support to demonstrate its overall performance. Please add more experiments on more real-world datasets, with comparisons to more cutting-edge forecasting methods.*
>
> **Response**: We have expanded the study on both axes.
>
> 1. *From two real-world datasets to a total of nineteen (19) real-world datasets.* The empirical study now spans datasets from areas namely dynamical systems in physics, ecology, hydrology, energy, environmental science, meteorology, transportation, astronomy and biology. See Appendix B.5 of the revised manuscript for the full list of datasets considered. Section 3.2 of the revised manuscript also has some details on these additions.
>
> 2. *Comparison with state-of-the-art competitors.* We have now expanded the competing methods considered and the new additions involve both classical statistical methods and also modern deep learning based methods.
>
> Among the competing methods listed in Figure 7 (and Table 12 in Appendix B) of the revised manuscript, the newly added classical statistical suite includes techniques such as ridge-regularized autoregression (ridge AR) [10] and  exponential smoothing (ETS) [11]. Among more  modern deep learning based forecasters, we consider N-BEATS [15], DeepAR [18], a compact Transformer [19], and N-HiTS [1]. We also list capacity-matched MLP and LSTM variants, whose model sizes are enlarged so their trainable-parameter count matches or exceeds the size of the selected AR-BPAR configuration. In Tables 11-14 in Appendix B of the revised manuscript, the capacity-matched variants have "(matched)" written. Forecasting performance of all methods are evaluated using the same  rolling one-step ahead protocol.
>
> Table 11 in the revised manuscript reports, per dataset, held-out test RMSE relative to the best method on that dataset. Table 12 shows the resulting overall ranking of different methods in terms of forecasting performance. In these tables, "MLP (matched)" and "LSTM (matched)" denote the capacity-matched variants: the same MLP/LSTM architectures re-tuned with hidden sizes enlarged so that their trainable parameter count matches or exceeds the selected BPAR/AR-BPAR model's configuration; see also  Appendix A.4 of the revised manuscript.  We see that across the 19 datasets, winners are spread over different methods. However, it must be noted that the new BPAR/AR-BPAR is consistently best-or-near-best in most cases. Table 12 in Appendix B reports the mean rank of the various methods along with the median relative test RMSE. Overall, we witness the strong performance of the AR-BPAR method. It must be emphasized that the new BPAR/AR-BPAR model operates at a small fraction of the parameter count used by the deep learning competitors; see the newly added Table 14 in Appendix B for the details on the competing methods' sizes.
>
> ---
> **References**
>
> [1] Cristian Challu, Kin G Olivares, Boris N Oreshkin, Federico Garza Ramirez, Max Mergenthaler Canseco, and Artur Dubrawski. N-hits: Neural hierarchical interpolation for time series forecasting. In Proceedings of the AAAI .
>
> [10] Arthur E Hoerl and Robert W Kennard. Ridge regression: Biased estimation for nonorthogonal problems. Technometrics, 12(1):55–67, 1970.
>
> [11] Rob J. Hyndman and George Athanasopoulos. Forecasting: Principles and Practice. OTexts, Melbourne, Australia, 3rd edition, 2021.
>
> [15] Boris N. Oreshkin, Dmitri Carpov, Nicolas Chapados, and Yoshua Bengio. N-beats: Neural basis expansion analysis for interpretable time series forecasting. In International Conference on Learning Representations, 2020.
>
> [18] David Salinas, Valentin Flunkert, Jan Gasthaus, and Tim Januschowski. Deepar: Probabilistic forecasting with autoregressive recurrent networks, 2017.
>
> [19] Ashish Vaswani, Noam Shazeer, Niki Parmar, Jakob Uszkoreit, Llion Jones, Aidan N Gomez, Lukasz Kaiser, and Illia Polosukhin. Attention is all you need. In Advances in Neural Information Processing Systems, volume 30, 201.

---

> ### Author Response · Authors · 2026-07-15
> **Requested change 2: Hyperparameter choices and discussion (part 1)**
>
> *While the paper describes how to search for the best hyperparameters, the empirical study section does not provide the resulting hyperparameter choices. Also, more discussion on the hyperparameters would be helpful to general readers.*
>
> **Response**: Table 10 in Appendix A.4 of the revised manuscript now reports the selected AR-BPAR configuration ($d_L$, $d_B$, $m$, input scaling function $h(\cdot)$, prior distribution parameter choices, and the resulting effective degrees of freedom) for every real-world dataset considered.
>
> Next, on the three specific question, we report a number of new results and related discussion.
>
> 1. *In general, how large a Bernstein polynomial degree is sufficient?*
>
> **Response**: Table 10 in Appendix A.4 of the revised manuscript reports the selected Bernstein polynomial degree across the 19 real-world datasets. We witness this polynomial degree range from $m=5$ to $m=40$. As an illustration with a specific chaotic time series dataset (Santa Fe laser), we report sensitivity analysis with respect to the polynomial degree $m$. Figure 34 in Appendix B of the revised manuscript shows that the forecasting RMSE attains near optimality $m\approx 10$–$20$,  and plateaus beyond it. We chose to present this particular dataset's result because we witnessed the highest amount of nonlinearity present with traditional linear time series models (ridge AR and ARIMA) struggling the most. Figure 36 contains the corresponding sensitivity analysis plots for the Rössler time series dataset, another highly nonlinear time series. The observations made with these two representative datasets were seen to broadly hold true for the other 17 real-world datasets considered in this article.
>
> Note that our Bayesian methodology in Section 2.3 of the manuscript places shrinkage priors on the Bernstein weight coefficients; see below Eq. (6) and Eq. (13) in the manuscript. This specifically penalizes selection of a large Bernstein polynomial degree $m$. This approach helps us strike a balance between model size and accuracy attained. Empirically this previous point is validated in the right plots of Figures 35 and 37 in Appendix B (Santa Fe laser time series and  Rössler time series).

---

> ### Author Response · Authors · 2026-07-15
> **Response to Requested change 2 (part 2)**
>
> 2. *What are the effects of different hyperparameter choices? For example, if a few hyperparameters are not chosen optimally, what would be the expected results?*
>
> **Response**: A mild degradation in test RMSE is what we witness, on average, across the 19 real-world datasets. See the newly added Appendix B.3. As an illustration, we consider the most nonlinear time series dataset from the list: Santa Fe laser. Figure 34 in Appendix B of the revised manuscript reveals that gain in RMSE is very minimal beyond a certain choice of the hyperparameters $d_L,d_B,m$. A similar story can be witnessed in the left plot in Figure 35 in Appendix B wherein we study sensitivity of RMSE as the $\tau_B$ parameter increases. Finally with the right plot in Figure 35, with $\tau_B$ fixed, we see that the effective degrees of freedom decreasing as $\gamma$ decreases. Similarly, Figures 36 and 37 contain the corresponding sensitivity analysis plots for the Rössler time series dataset. The observations made with these two representative datasets were seen to broadly hold true for the other 17 real-world datasets considered in this article.
>
> 3. *Is there any trade-off for different hyperparameter choices? For example, does a large Bernstein polynomial degree imply higher computation cost to evaluate? Is there any trade-off between accuracy and efficiency?*
>
> **Response**:  The exact Bayesian inference method in Section 2.3.2 of the manuscript, the computation involves $K_{\theta}\times K_{\theta}$ sized matrix operations, and the associated
> cost grows polynomial with respect to the Bernstein polynomial degree $m$. Across the 19 real-world datasets considered in Section 3.2, the model size is typically  $K_{\theta} \le 300$. The exact Bayesian inference from Section 2.3.2 takes at most 10 milliseconds. Further, the complete hyperparameter search involving the technique from Section 2.4 takes a  median time of 9 seconds (maximum 31 seconds) per dataset on a single CPU core, with no GPU required. In comparison, the variational Bayesian inference route from Section 2.3.1 of the manuscript, estimated via backpropagation, is slower: a single model fit takes about 2 seconds on average, and the corresponding complete hyperparameter search takes a median time of approximately 2,060 seconds (maximum approximately 4,040 seconds) per dataset on the same single CPU core.
>
> Table 14 shows that even the smallest model  configuration among the neural network/deep learning competitors is larger than our AR-BPAR's size, and this underscores the parsimony of the proposed model without compromising too much on test RMSE. Figure 33 shows the resulting accuracy–size frontier: the proposed BPAR/AR-BPAR models reach top-tier accuracy at parameter counts that are orders of magnitude below the larger neural network/deep learning baselines.
>
> We thank all Reviewers again for the detailed and insightful comments.

---

### Review · Reviewer_ByTS · 2026-06-23

**Summary Of Contributions:**

**Disclaimer**: *my prior background in ML is vision, language, speech processing which is different from the scope of the paper, however I hold PhD in math and can judge on probability theory, optimization, functions approximation theory, etc. Thus I might not be closely familiar with the prior literature on the matter of the paper.*

The paper proposes to model non-linear time series as a linear model on the Bernstein polynomials (they are used as a basis). Empirically, authors show that this simple model (as it is interpretable) can model various non linear time series consistently being similar or better than prior methods in approximation quality. Also authors show how Bayesian inference (including variational) can be applied to get confidence intervals - necessary for application features which some NN-based methods lack.

**Strengths**
- simple interpretable model which is based on approximation theory of Bernstein polynomials
- empirical results which show that method is on par or better (most of the time marginal) than prior methods (but prior methods vary which one is better depending on the data) -> so the proposed method seems to be robust and consistently better / same for all considered data.
- the paper is well written overall, simple to read, nice plots in appendix showing how different models approximate the considered time series.

**Weaknesses**
- Authors consider Bayesian inference as a contribution, however math itself is standard ELBO derivation under simple model as well as classic statistics for the non-variational part. There is no estimation of confidence intervals on empirical data showing that these intervals are reflecting reality (that are reliable actually) and there is no comparison with prior methods which can produce confidence intervals showing that authors' approximation now provides more accurate confidence intervals. Thus the whole part of the Bayesian inference looks standalone and not tight to the paper. I really feel that I am missing a point, even why it is important to have? Why is it critical? I am not sure that derivations themselves are for publication as this is a simple university exercise given that the reparametrization trick is used and standard statistical methods are used too. Maybe contribution is consideration of specific prior - but then there should be some discussion / comparison on that too to justify it (though from math and modeling point it looks reasonable).
- As I am not too familiar with the literature I checked a bit if Bernstein polynomials were used before, so I found this work "Lukoseviciute, K., Baubliene, R., Howard, D. and Ragulskis, M., 2018. Bernstein polynomials for adaptive evolutionary prediction of short-term time series. Applied Soft Computing, 65, pp.47-57." which from the first glance seems to be using a similar idea but for short time horizon prediction. Can authors comment on this work and compare it with the current paper?
- There are many approximation theorems and basis functions, including different polynomial families - why exactly Bernstein - this is not explained in the paper, thus a bit questions the underlying reasoning of the method. Maybe we can go even bigger and say that any polynomial family can be used for modeling and it will be better than MLP / LSTM or other methods?
- Authors stress the importance of interpretability - there should be some references or more discussion why do we care about it in the end and how it is used in application? Maybe I am happy with black-box model but which produces an almost exact approximation?
- There are no ablations on a bunch of hyperparameters to show robustness or to show complexity of the method.

**Audience:**

Yes

**Audience Explanation:**

Results on approximation of time series could be of interest to TMLR community, though the amount of people will be limited as this covers very narrow applications - the method cannot be expanded to other domains or input data type.

**Broader Impact Concerns:**

No issues.

**Claims And Evidence:**

No

**Claims Explanation:**

- Bayesian inference is not supported with justification why it is needed, why it is better than prior methods, what intervals it gives for real data. Math on itself seems to be very classic, so I am not sure if I can consider this as a theoretical contribution.
- It is not explained why to use exactly Bernstein polynomials, moreover as listed above there is prior work which used them already for time series modeling
- There is no ablations on the model to show its robustness or compare with prior works for the same function complexity (e.g. maybe results are better because the model is just bigger as larger order and larger history lag is used)

**Requested Changes:**

First of all I would like to see responses to listed weaknesses and changes should address them.

Second, I list some minor issues with the writing / presentation of the work:
- (personal, so can be ignored) I am not sure it is nice to compare with single layer perceptron - is it good that it is so simple? Or is it a different message? I was literally reading that a single layer perceptron is a strong model and we are using it. Be precise why it is good / bad to be like a single layer perceptron and why.
- Add info why interpretability is important (even maybe crucial?), as otherwise I care about the more performance and that it can solve my problem despite interpretability (but maybe my problem solving is exactly being interpretable? - you need to specify this).
- "Carefully designed shrinkage prior distributions are utilized in the Bayesian inference, and they assist in maintaining overall model parsimony." - I don't understand the meaning of "model parsimony" and to which exactly it is referencing.
- "unique ability of BPAR to provide uncertainty quantification in these data settings." In the text it is not explained why prior methods cannot, why it is important (some reference), as well as validation that estimated intervals are meaningful and helpful (it can be that uncertainty will be large, thus useless in practice).
- Paragraph after "The main contributions are summarized as follows." repeats exactly the previous paragraph - I would suggest removing repetition and reframing a bit.
- "sequence of random variables ${z_t}$" I didn't get why z is random variable? Also, what is $h(z)$ function? what form for it did you use?
- There is no discussion how $d_L$, $d_B$, $m$ affects the approximation quality (point on hyperparameters ablations). There is unclear comparison between prior methods model sizes and the proposed method. Also I wonder if you use another polynomial family will it be helpful the same way? Kind of answering why Bernstein exactly.
- There is no discussion about what MLP and LSTM models are used, is capacity in these models enough? etc. Or some reference to prior works where you matched SOTA results and I can trust the numbers on the baselines. Same goes to AR models e.g. (Sorry, I am not familiar with the literature on SOTA models, so maybe that is why it is a naive question from my side).
- Section 3.1 ending - the key point for me was actually that your method works across the board while for prior works some are the best on one data but worse on others. Maybe highlight this point instead? I am also curious if you used exactly the same $d_L$, $d_B$, $m$  for all listed data?
- Figure 8 in Appendix - it seems to be there is a constant shift in all approximations in time, why is that?
- Looking at chaotic time series approximations, it seems that all prior works cannot model high frequencies in the signal, but does your method can do it because you use high order of the polynomials? technically then I need to have high non-linearity in MLP e.g. using more layers, so did you try more complicated / deeper models (still staying with a small number of parameters as data are limited)?

---

> ### Author Response · Authors · 2026-07-15
> **Weakness 1: Bayesian inference and its validation (part 1)**
>
> *Authors consider Bayesian inference as a contribution, however math itself is standard ELBO derivation under simple model as well as classic statistics for the non-variational part. There is no estimation of confidence intervals on empirical data showing that these intervals are reflecting reality (that are reliable actually) and there is no comparison with prior methods which can produce confidence intervals showing that authors' approximation now provides more accurate confidence intervals. Thus the whole part of the Bayesian inference looks standalone and not tight to the paper. I really feel that I am missing a point, even why it is important to have? Why is it critical? I am not sure that derivations themselves are for publication as this is a simple university exercise given that the reparametrization trick is used and standard statistical methods are used too. Maybe contribution is consideration of specific prior — but then there should be some discussion / comparison on that too to justify it (though from math and modeling point it looks reasonable).*
>
> **Response**:
> The key contribution of the Bayesian treatment is the design of the lag- and Bernstein-degree-dependent shrinkage prior distributions; see below Eq. (6) and Eq. (13) in the manuscript. Observe that the prior mean of each Bernstein weight coefficient is zero, and the prior variance decays with respect to the time lag index and the polynomial degree index. These priors help  deliver a form of model parsimony—principled, automatic pruning of high-lag and high-degree terms via posterior credible intervals. Specifically, posterior credible intervals can be utilized to identify the statistically significant and non-significant model parameters.  The Bayesian treatment is thus a means to two ends we do claim: (a) this shrinkage-based model parsimony, and (b) calibrated predictive intervals for chaotic and highly nonlinear series. In the revised manuscript, we have clarified the language in Section 1, and also at the beginning of Section 2.3.

---

> ### Author Response · Authors · 2026-07-15
> **Response to Weakness 1 (part 2)**
>
> **Response to Weakness 1 (continued).** Next, we mention the new empirical evidence regarding the use of posterior predictive intervals.
>
> In Appendix B.1 of the revised manuscript, we discuss  certain measures used for assessing the quality of prediction intervals obtained by applying the Bayesian procedure from Section 2.3.2 of the manuscript. First, we consider Prediction interval coverage probability ($PICP$) that measures the fraction of test points that the interval captures. A well-calibrated model has $PICP(\alpha)\approx 1-\alpha$; see [12] for details. Second, the mean relative prediction interval width ($MPIW_{rel}$) from  [12] which speaks to the sharpness of the interval. $PICP$ and $MPIW_{rel}$ must be read together: coverage is only meaningful at a stated sharpness, since an arbitrarily wide interval attains $PICP=1$ trivially. Third, the continuous ranked probability score (CRPS) that assesses the entire predictive distribution against the realized value; see [6] for details.
>
> Table 13 in Appendix B.1  reports, per dataset, the PICP at the coverage $90\%$ and $95\%$ levels, the relative MPIW, and the CRPS of AR-BPAR's closed-form Student-$t$ intervals.  Note that for BPAR/AR-BPAR models, these metrics evaluate the exact Student-$t$ posterior predictive intervals; see Section 2.3.2. Across the 19 real-world datasets, AR-BPAR's $90\%$ intervals attain a median empirical coverage of $0.92$.
>
> In that table, only  DeepAR's CRPS is displayed alongside for comparison because DeepAR is a competing method with a native predictive distribution at comparable point accuracy, and also is the strongest probabilistic baseline in the study (best CRPS on 13 of 19 datasets). Hence, it provides the most demanding like-for-like reference for BPAR/AR-BPAR's interval quality.  ARIMA/SARIMA analytic intervals are summarized in the table footer. DeepAR's intervals are computed as follows: the model is trained with a Gaussian likelihood on the network output, and its predictive distribution at each test step is obtained by ancestral sampling of full trajectories from the fitted model [18], as implemented in the `darts` package [8]; we draw $500$ sample paths per rolling one-step forecast, summarize them by their sample mean and standard deviation, form the $90\%$/$95\%$ intervals from the corresponding Gaussian quantiles, and evaluate the CRPS with the closed-form expression for a Gaussian predictive [6].
> We see that BPAR/AR-BPAR attains the best CRPS at 6 of 19 datasets, but  delivers a performance that is highly competitive with DeepAR. It must be emphasized that the new BPAR/AR-BPAR model operates at a small fraction of the parameter count used by DeepAR; see the newly added Table 14 in Appendix B for the exact counts.
>
> Finally, as an illustration, we consider two highly nonlinear time series datasets from our list (Santa Fe laser and Rössler) and plot the nominal and empirical coverage of the posterior predictive intervals; see Figure 32 in Appendix B.1 of the revised manuscript.
>
> ---
> **References**
>
> [6] Tilmann Gneiting and Adrian E Raftery. Strictly proper scoring rules, prediction, and estimation. Journal of the American Statistical Association, 102(477):359–378, 2007.
>
> [8] Julien Herzen, Francesco L"assig, Samuele Giuliano Piazzetta, Thomas Neuer, L'eo Tafti, Guillaume Raille, Tomas Van Pottelbergh, Marek Pasieka, Andrzej Skrodzki, Nicolas Huguenin, et al. Darts: User-friendly modern machi.
>
> [12] Abbas Khosravi, Saeid Nahavandi, Doug Creighton, and Amir F Atiya. Comprehensive review of neural network-based prediction intervals and new advances. IEEE Transactions on Neural Networks, 22(9):1341–1356, 2011.
>
> [18] David Salinas, Valentin Flunkert, Jan Gasthaus, and Tim Januschowski. Deepar: Probabilistic forecasting with autoregressive recurrent networks, 2017.

---

> ### Author Response · Authors · 2026-07-15
> **Weakness 2: Prior work using Bernstein polynomials**
>
> *As I am not too familiar with the literature I checked a bit if Bernstein polynomials were used before, so I found this work "Lukoseviciute, K., Baubliene, R., Howard, D. and Ragulskis, M., 2018. Bernstein polynomials for adaptive evolutionary prediction of short-term time series. Applied Soft Computing, 65, pp.47-57." which from the first glance seems to be using a similar idea but for short time horizon prediction. Can authors comment on this work and compare it with the current paper?*
>
> **Response**: Bernstein polynomials have been used to model the trajectory of time series; see [14] for an example. There, the Bernstein polynomial is an interpolating curve in the time index and forecasts are obtained by extrapolating that curve. Specifically, Eq. (12) of their work provides the extrapolation equation, and structurally, this points to modeling linear temporal autocorrelations between past values of the time series and the present/future.
> In contrast, our BPAR setup models nonlinear temporal autocorrelations using Bernstein polynomials. There is a structural difference to the earlier cited work. Specifically,  the nonlinear influence of each time-lagged observation on the current state of the time series is modeled via Bernstein polynomials. This allows more flexibility to capture highly nonlinear temporal dependence structures, as often seen in chaotic time series. We have made this clarification in a newly added Remark 2.2 in Section 2 of the revised manuscript.
>
> Our BPAR model equation (Eq. (2) of the manuscript) resembles a nonparametric regression equation with the Bernstein polynomials serving as the nonlinear regression function. Bernstein polynomials have been used in a number of nonparametric regression related works; see [2, 16, 20]. Keep in mind that these works are not setup in the time series context.  There again, the general modeling assumption is that a response variable $Y_i$ at observation index $i$ can be expressed as $Y_i = f(X_i) + \textrm{error}$, where $X_i$ is the explanatory variable. The function $f(\cdot)$ is then assumed to be smooth and approximated via Bernstein polynomials. See also response to the immediately following Weakness critique.
>
> In the time series context,  prediction intervals using nonlinear autoregressive models have been developed and studied in the statistical literature; see [4, 17, 21] for examples. The model assumption in these works is that $x_t = m(x_{t-1},x_{t-2},\cdots,m_{t-p}) + \epsilon_t$, for an order $p$ model wherein $m(\cdot)$ is a possibly nonlinear function. If $m(\cdot)$ is assumed to be smooth, the problem then becomes a nonparametric autoregressive model with $m(\cdot)$ being estimated using kernel methods. In practice, when implementing their methods,  $m(\cdot)$ is estimated using kernels or assumed to be known. Our BPAR model is more flexible than the above described setup as it models the nonlinear influence of each time-lagged observation. As evidenced by the results in our empirical study, this previous point is seen to be true when dealing with chaotic time series. Further, the Bernstein polynomials help avoid the difficulty in selecting the unknown $m(\cdot)$ function; see Remark 2.1 in the manuscript.
>
> Hence the newly proposed models in Sections 2 of the manuscript is structurally different from the literature mentioned above.
>
> ---
> **References**
>
> [2] S. M. Curtis and S. K. Ghosh. A variable selection approach to monotonic regression with bernstein polynomials. Journal of Applied Statistics, 38:961–976, 2011.
>
> [4] J"urgen Franke, Jens-Peter Kreiss, and Enno Mammen. Bootstrap of kernel smoothing in nonlinear time series. Bernoulli, 8(1):1–37, 2002.
>
> [14] Kristina Lukoseviciute, Rita Baubliene, Daniel Howard, and Minvydas Ragulskis. Bernstein polynomials for adaptive evolutionary prediction of short-term time series. Applied Soft Computing, 65:47–57, 2018.
>
> [16] M. Osman and S. K. Ghosh. Nonparametric regression models for right-censored data using bernstein polynomials. Computational Statistics & Data Analysis, 56:559–573, 2012.
>
> [17] Li Pan and Dimitris N. Politis. Bootstrap prediction intervals for linear, nonlinear and nonparametric autoregressions. Journal of Statistical Planning and Inference, 177:1–27, 2016.
>
> [20] J. Wang and S. K. Ghosh. Shape restricted nonparametric regression with bernstein polynomials. Computational Statistics & Data Analysis, 56:2729–2741, 2012.
>
> [21] Kejin Wu and Dimitris N. Politis. Bootstrap prediction inference of nonlinear autoregressive models. Journal of Time Series Analysis, 45(5):800–822, 2024. endthebibliography.

---

> ### Author Response · Authors · 2026-07-15
> **Weakness 3: Why exactly Bernstein polynomials?**
>
> *There are many approximation theorems and basis functions, including different polynomial families — why exactly Bernstein — this is not explained in the paper, thus a bit questions the underlying reasoning of the method. Maybe we can go even bigger and say that any polynomial family can be used for modeling and it will be better than MLP / LSTM or other methods?*
>
> **Response**: The BPAR/AR-BPAR model proposed in Section 2 of the manuscript has structural resemblance to the nonparametric regression model. Bernstein polynomials have been used in a number of nonparametric regression related works; see [2, 16, 20]. In the non-time series context, a response variable $Y_i$ at observation index $i$ is expressed as $Y_i = g(X_i) + \textrm{error}$, where $X_i$ is the explanatory variable. The function $g(\cdot)$ is then assumed to be continuous and can be approximated via Bernstein polynomials. [16], under the nonparametric regression setup, discuss certain key advantages of using Bernstein polynomials over any other polynomial basis. First, statistical consistency properties have been established with rate of convergence similar to the usual kernel type estimators of the regression function. Second, it has been shown that in order to establish these asymptotic theoretical guarantees, high-order smoothness assumptions are not required, but mere continuity is adequate. Third, Bernstein polynomials cater well to shape restrictions in the regression function: for example monotone, convex, concave, etc... Though beyond the scope of this current work, the theoretical results from the previous cited work, can be extended to the time series setting that we have with the BPAR model, under certain regularity assumptions concerning the cumulants of the observed time series.  A new Remark 2.3 is added in Section 2 of the revised manuscript.
>
> ---
> **References**
>
> [2] S. M. Curtis and S. K. Ghosh. A variable selection approach to monotonic regression with bernstein polynomials. Journal of Applied Statistics, 38:961–976, 2011.
>
> [16] M. Osman and S. K. Ghosh. Nonparametric regression models for right-censored data using bernstein polynomials. Computational Statistics & Data Analysis, 56:559–573, 2012.
>
> [20] J. Wang and S. K. Ghosh. Shape restricted nonparametric regression with bernstein polynomials. Computational Statistics & Data Analysis, 56:2729–2741, 2012.

---

> ### Author Response · Authors · 2026-07-15
> **Weakness 4: Why interpretability matters**
>
> *Authors stress the importance of interpretability — there should be some references or more discussion why do we care about it in the end and how it is used in application? Maybe I am happy with black-box model but which produces an almost exact approximation?*
>
> **Response**: With time series data, in addition to providing uncertainty quantification on predictions, being able to statistically test for nonlinearity in temporal autocorrelations, long memory and statistical significance of model parameters is of immense use in application areas such as economics and finance [3, 7, 5]. These works are viewed as a hybrid between classical time series models and artificial neural network models and they employ techniques such as bootstrapping, Bayesian inference to conduct statistical tests. A few sentences have now been added to Section 1 of the revised manuscript.  Next, we include some model interpretability-related discussion concerning one of the real-world datasets we have considered in this article: Ontario electricity price time series.
>
> We have also added a new section Appendix A.5. There we consider two real-world datasets: Ontario electricity price time series and Beijing PM2.5 time series. We illustrate the use of posterior credible intervals on the Bernstein weight coefficients in detecting for presence/absence of nonlinear temporal dependence in the two time series. The credible intervals show that there is significant nonlinearity present in the Ontario electricity price time series, whereas the Beijing PM2.5 time series did not reveal significant nonlinearity in temporal dependence. This result can also be connected to the test RMSE results in Table 11 in Appendix B of the revised manuscript. There we see that for the Ontario electricity price time series, the drop in performance from the linear ARIMA model when compared to the better performing AR-BPAR model. On the contrary, with the Beijing PM2.5 time series, due to the absence of significant nonlinearity, the linear ARIMA and the new AR-BPAR model show very close performance.
>
> ---
> **References**
>
> [3] Ersan Eğrioğlu and Robert Fildes. A new bootstrapped hybrid artificial neural network approach for time series forecasting. Computational Economics, 59:1355–1383, 2022.
>
> [5] Fumitaka Furuoka, Luis A. Gil-Alana, OlaOluwa S. Yaya, Elayaraja Aruchunan, and Ahamuefula E. Ogbonna. A new fractional integration approach based on neural network nonlinearity with an application to testing unemploymen.
>
> [7] Niko Hauzenberger, Florian Huber, Karin Klieber, and Massimiliano Marcellino. Bayesian neural networks for macroeconomic analysis. Journal of Applied Econometrics, 2024. Preprint available on arXiv.

---

> ### Author Response · Authors · 2026-07-15
> **Weakness 5: Hyperparameter ablations and model complexity**
>
> a. The Bernstein polynomial degree $m$: Table 10 in Appendix A.4 of the revised manuscript reports the selected Bernstein polynomial degree across the 19 real-world datasets. We witness this polynomial degree range from $m=5$ to $m=40$. As an illustration with a specific chaotic time series dataset (Santa Fe laser), we report sensitivity analysis with respect to the polynomial degree $m$. Figure 34 in Appendix B of the revised manuscript shows that the forecasting RMSE attains near optimality $m\approx 10$–$20$,  and plateaus beyond it. We chose to present this particular dataset's result because we witnessed the highest amount of nonlinearity present with traditional linear time series models (ridge AR and ARIMA) struggling the most. Figure 36 contains the corresponding sensitivity analysis plots for the Rössler time series dataset, another highly nonlinear time series. The observations made with these two representative datasets were seen to broadly hold true for the other 17 real-world datasets considered in this article.
>
> b. Hyperparameter sensitivty analysis: Please see the newly added Appendix B.3 for details.
>
> c. Hyperparameter choice/size vs accuracy vs computing cost: With the exact Bayesian inference method in Section 2.3.2 of the manuscript, the computation involves $K_{\theta}\times K_{\theta}$ sized matrix operations, and the associated
> cost grows polynomial with respect to the Bernstein polynomial degree $m$. Across the 19 real-world datasets considered in this article, the model size is seen to be $K_{\theta} \le 300$. The exact Bayesian inference method from Section 2.3.2 takes at most 10 milliseconds. Further, the complete hyperparameter search involving the technique from Section 2.4 takes a  median time of 9 seconds (maximum 31 seconds) per dataset on a single CPU core, with no GPU required. In comparison, the variational Bayesian inference route from Section 2.3.1 of the manuscript, estimated via backpropagation, is slower. A single model fit takes about 2 seconds on average, and the corresponding complete hyperparameter search takes a median time of approximately 2{,}060 seconds (maximum approximately 4{,}040 seconds) per dataset on the same single CPU core.
>
> Table 14 shows that even the smallest model  configuration among the neural network/deep learning competitors is larger than our AR-BPAR's model size, and this underscores the parsimony of the proposed model without compromising too much on prediction accuracy. Figure 33 shows the resulting accuracy–size frontier. There, we see that the proposed BPAR/AR-BPAR models reach top-tier accuracy at parameter counts that are orders of magnitude below the larger neural network/deep learning competitors. This discussion is added to Appendix B.4 of the revised manuscript.

---

> ### Author Response · Authors · 2026-07-15
> **Minor issue 1**
>
> > *(personal, so can be ignored) I am not sure it is nice to compare with single layer perceptron — is it good that it is so simple? Or is it a different message? I was literally reading that a single layer perceptron is a strong model and we are using it. Be precise why it is good / bad to be like a single layer perceptron and why.*
>
> **Response**: We have removed that specific MLP comparison sentence from the abstract. The key point we wish to convey is that the new BPAR model architecturally resembles a single-layer artificial neural network with the Bernstein polynomials playing the role of pseudo-activation functions. This route avoids selecting activation functions and brings interpretable weight coefficients using Bayesian inference. This route also enable subsequent tests on importance of certain time lags for prediction and  presence/absence of nonlinear temporal dependence in the data.

---

> ### Author Response · Authors · 2026-07-15
> **Minor issue 2**
>
> > *Add info why interpretability is important (even maybe crucial?), as otherwise I care about the more performance and that it can solve my problem despite interpretability (but maybe my problem solving is exactly being interpretable? — you need to specify this).*
>
> **Response**: Please see the response to Weakness 4 above.

---

> ### Author Response · Authors · 2026-07-15
> **Minor issue 3**
>
> > *"Carefully designed shrinkage prior distributions are utilized in the Bayesian inference, and they assist in maintaining overall model parsimony." — I don't understand the meaning of "model parsimony" and to which exactly it is referencing.*
>
> **Response**: Model parsimony means being able to work with fewer model parameters without compromising too much on prediction accuracy. Specifically, in Section  2.3, we use the notation $K_{\theta}$ to denote the size of the model's parameter vector. With the shrinkage prior distributions penalizing high time lag and high Bernstein polynomial degree selections, posterior credible intervals can be used to prune the model to only include the statistically significant coefficients. The sentence in the Abstract is reworded.

---

> ### Author Response · Authors · 2026-07-15
> **Minor issue 4**
>
> > *"unique ability of BPAR to provide uncertainty quantification in these data settings." In the text it is not explained why prior methods cannot, why it is important (some reference), as well as validation that estimated intervals are meaningful and helpful (it can be that uncertainty will be large, thus useless in practice).*
>
> **Response**: We have now reworded the abstract to not mention "unique". We intend to convey that our work performs better than traditional, mostly linear, time series models in terms of prediction accuracy when analyzing highly nonlinear time series datasets. At the same time, we offer, via posterior credible intervals, uncertainty quantification on the model parameters and the predictions.
>
> Please also see the newly added Appendix B.1 that provides empirical evidence on the performance of the posterior predictive intervals. Please also see newly added Appendix A.5 that illustrates the use of posterior credible intervals on model parameters to detect for presence/absence of nonlinear temporal autocorrelations in the data.

---

> ### Author Response · Authors · 2026-07-15
> **Minor issue 5**
>
> > *Paragraph after "The main contributions are summarized as follows." repeats exactly the previous paragraph — I would suggest removing repetition and reframing a bit.*
>
> **Response**: Fixed.

---

> ### Author Response · Authors · 2026-07-15
> **Minor issue 6**
>
> > *"sequence of random variables $z_t$" — I didn't get why $z$ is a random variable? Also, what is the $h$ function? What form for it did you use?*
>
> **Response**: Function $h(\cdot)$ is a deterministic, injective transform of the time series $x_t$ onto the interval $[0,1]$. This means that with $x_t$ assumed to be a sequence of random variables, $z_t = h(x_t)$ is also a sequence of random variables. In practice, we work with two choices for $h$. First, the  min–max scaling $h(x) = (x - \min_{x})/(\max_{x} - \min_{x})$, $\max_{x}$ and $\min_{\mathrm{tr}}$ are assumed to be known maximum and minimum values attained by the time series. Second, the logistic-sigmoid function $h(x) = 1/(1+e^{-x})$. Some additional clarity is added below Eq. (2) of the revised manuscript.
>
> To get more technically precise, let $F_{x,t} = \sigma(x_t,x_{t-1},\cdots)$ be the sigma-algebra spanned by the sequence of random variables $x_t$. Then, with the injective map $h(\cdot)$, $F_{z,t} \subset F_{x,t}$, i.e., the sigma-algebra of the sequence of random variables $z_t$ is a sub-sigma-algebra of $F_{x,t}$. This holds as long as $h(\cdot)$ is a measurable function.

---

> ### Author Response · Authors · 2026-07-15
> **Minor issue 7**
>
> > *There is no discussion how $d_L$, $d_B$, $m$ affect the approximation quality (point on hyperparameters ablations). There is unclear comparison between prior methods' model sizes and the proposed method. Also I wonder if you use another polynomial family will it be helpful the same way? Kind of answering why Bernstein exactly.*
>
> **Response**: Please refer to our response to Weakness 5 mentioned above. Sensitivity analysis with respect to various hypermarameters and details on competing methods' sizes and how they are selected are now added to Appendix B. Specifically, please look at Appendix A.4, B, B.2, B.3, B.4.
>
> Regarding the question of "why specifically Bernstein polynomials?", please refer to our response to Weaknesses 2 and 3 above.

---

> ### Author Response · Authors · 2026-07-15
> **Minor issue 8**
>
> > *There is no discussion about what MLP and LSTM models are used, is capacity in these models enough? etc. Or some reference to prior works where you matched SOTA results and I can trust the numbers on the baselines. Same goes to AR models e.g.*
>
> **Response**:
> The criteria used to select the MLP and LSTM configurations are now specified in Appendix A.4 of the revised manuscript. In brief: for the MLP, a grid search over the input lag order in $\{3,6,12,24\}$, hidden-layer architectures in $\{(16),(32),(32,16)\}$ including a two-layer option, and $\ell_2$ penalty in $\{10^{-4},10^{-3}\}$; each candidate uses ReLU activations and is trained by Adam with validation-based early stopping. For the LSTM, a grid search over the input sequence length in $\{12,24,48\}$, hidden state size in $\{8,16\}$, and learning rate in $\{10^{-3},3\times 10^{-4}\}$; each candidate is a single-layer LSTM with a linear head, trained by the Adam algorithm [13]. In both cases the configuration attaining the lowest validation RMSE under the same chronological train/validation protocol is selected, and every comparison table reports trainable-parameter counts. Regarding whether the capacity of these models is enough: the capacity-matched MLP/LSTM variants repeat the identical grid search above with hidden sizes enlarged to push width/depth further so that their trainable-parameter counts match or exceed(mostly) the selected AR-BPAR's on each dataset (Table 14 in Appendix B.2), and the modern deep learning baselines (N-BEATS, DeepAR, Transformer, N-HiTS) are strong on several datasets, so the baseline suite is demonstrably not underpowered. The same selection principle applies to the linear and classical baselines. For the ridge AR model, we perform a grid search over the autoregressive lag order in $\{1,3,6,12,24\}$ and the ridge penalty in $\{0.1,\,1,\,10\}$; each candidate is estimated in closed form on the training window from the lagged design matrix. For the ARIMA model, we perform a grid search over the orders $p\in\{0,1,2,3\}$, $d\in\{0,1\}$, and $q\in\{0,1,2\}$, with each candidate estimated by maximum likelihood using the `statsmodels` state-space implementation; the SARIMA model additionally searches the seasonal orders $(P,D,Q)\in\{0,1\}^{3}$ at the dataset's seasonal period (the seasonal component is omitted when the training window contains too few seasonal cycles for it to be identifiable). In all cases the configuration attaining the lowest validation RMSE under the same chronological train/validation protocol is selected—the identical criterion used for the MLP, LSTM, and our own BPAR/AR-BPAR models—so no method is favored by its selection rule; the selected ARIMA/SARIMA model is then re-estimated on the combined training and validation data before test-set forecasting, and an estimation-convergence check is recorded for every selected model.
>
> *Location in revised manuscript:* Appendix A.4 (selection criteria for the MLP and LSTM); Appendix B.2 (Table 14).
>
> ---
> **References**
>
> [13] Diederik P. Kingma and Jimmy Ba. ADAM: A Method for Stochastic Optimization. In *Proceedings of the 3rd International Conference on Learning Representations (ICLR)*, San Diego, CA, USA, 2015. arXiv:1412.6980, 2017 revision. DOI: 10.48550/arXiv.1412.6980.

---

> ### Author Response · Authors · 2026-07-15
> **Minor issue 9**
>
> > *Section 3.1 ending — the key point for me was actually that your method works across the board while for prior works some are the best on one data but worse on others. Maybe highlight this point instead? I am also curious if you used exactly the same $d_L$, $d_B$, $m$ for all listed data?*
>
> **Response**: Please see the new paragraph added right before the start of Section 4.

---

> ### Author Response · Authors · 2026-07-15
> **Minor issue 10**
>
> > *Figure 8 in Appendix — it seems to be there is a constant shift in all approximations in time, why is that?*
>
> **Response**: We investigated the indexing first and verified that each forecast is plotted against the time index it predicts: for a lag order $p$, each supervised row uses $(X_{t-1},\ldots,X_{t-p})$ as its input and $X_t$ as its target, the held-out test targets are the time indices $240$–$299$, and the observed targets and all forecasts are plotted against this same test-index vector. Figure 8 (Figure 9 in revised manuscript) therefore contains no off-by-one plotting error.
>
> The apparent one-step delay is instead a well-known visual property of one-step-ahead conditional-mean forecasting on series with strong time lag-1 dependence;
> see [9] for a discussion of exactly this phenomenon as a common pitfall in forecast evaluation, and [11] for the use of the naive (last-value) forecast as the reference benchmark on such strongly persistent series. This phenomenon is documented specifically in Section 3.1.1 of [9]: on a unit-root (near-random-walk) time series, the one-step forecasts of all the trained models in their Figure 5 behave like the naïve forecast, which "is a shifted version of the time series where the forecast simply follows the actuals" (their Fig. 5b), and their Figure 7 repeats the observation on a random walk series. Their Section 3.4 ("Forecast plots") further cautions that concluding models fit well from overlaid forecast plots "can lead to wrong conclusions"—the principled checks are benchmark comparisons and residual diagnostics rather than visual inspection. In the same spirit, Section 5.2 of [11] defines the naive (last-value) forecast—exactly the observed series shifted by one step—and notes that it is the optimal forecast when the data follow a random walk, which is why it serves as the reference benchmark on strongly persistent series; and their Section 5.3 recommends checking a fitted model through its residuals, i.e., whether it "has adequately captured the information in the data"—precisely the role of the residual Ljung–Box diagnostics used in our selection procedure (Section 2.4 of the manuscript) and reported alongside the accuracy metrics in Table 1.
>
> Because Models A–C are simulated, we can make this precise: the exact conditional mean $E[X_t \mid X_{t-1}]$ is known in closed form. This oracle predictor, which involves no estimation at all, exhibits the same one-step shift—on Model A its correlation with the previous observation $X_{t-1}$ is $0.98$, versus $0.76$ with the value $X_t$ it actually predicts—and our fitted models attain test RMSEs within a few percent of the oracle's ($0.591$ vs. $0.543$ on Model A, $0.212$ vs. $0.209$ on Model B, $0.478$ vs. $0.469$ on Model C). The shift is therefore a property of *optimal* one-step forecasting for these noise-driven processes, not an artifact: the innovation accounts for $66$–$81\%$ of each step's movement even under the true data-generating equations, and this unpredictable component can only be echoed one step later. The same reasoning explains why no such shift is visible in our other forecast overlays, for example the Mackey–Glass panels: those benchmarks are nearly deterministic, with the one-step-unpredictable component amounting to only $3$–$5\%$ of the series' standard deviation, so the conditional mean reproduces the trajectory at the correct index (their horizontal axes are segment-relative indices, a labeling convention that does not affect the target/forecast alignment). Finally, we note that forcing the forecast curve to visually overlap the observations—by shifting the held-out forecasts by one index—would evaluate the forecast for time $t+1$ against the observation at time $t$, and would no longer represent the stated forecasting task.
>
> ---
> **References**
>
> [9] Hansika Hewamalage, Klaus Ackermann, and Christoph Bergmeir. Forecast evaluation for data scientists: common pitfalls and best practices. Data Mining and Knowledge Discovery, 37(2):788–832, 2023.
>
> [11] Rob J. Hyndman and George Athanasopoulos. Forecasting: Principles and Practice. OTexts, Melbourne, Australia, 3rd edition, 2021.

---

> ### Author Response · Authors · 2026-07-15
> **Minor issue 11**
>
> > *Looking at chaotic time series approximations, it seems that all prior works cannot model high frequencies in the signal, but does your method can do it because you use high order of the polynomials? Technically then I need to have high non-linearity in MLP e.g. using more layers, so did you try more complicated / deeper models (still staying with a small number of parameters as data are limited)?*
>
> **Response**:  The answer to this question mostly rests in Remark 2.1 of the manuscript. In nonlinear autoregressive models, the typical assumption is that $x_t = m(x_{t-1},x_{t-2},\cdots,x_{t-p}) + \epsilon_t$, for an order $p$ model wherein $m(\cdot)$ is a possibly nonlinear function and $\epsilon_t$ is the error term. One way to understand high frequency oscillations in time series data is when there is strong negative autocorrelation between observations at  successive time points. For instance, $x_{t-1}$ strongly negatively  influences $x_t$ in a highly nonlinear manner.  With that point in mind, we argue that our BPAR model is much more flexible than the above described setup as it models the nonlinear influence of each time-lagged version of the time series.  This largely explains the better performance of the new models particularly with chaotic time series data.
>
> Now, with the MLP, one then needs to grow the model configuration (layers and nodes) in a highly targeted manner to be able to successfully capture this nonlinear negative impact of $x_{t-1}$ on $x_t$. Deeper layers and nodes could potentially help capture high frequency oscillations, but growing such a specific model architecture, with frequency oscillations in mind, is not very straightforward and needs further thought.

---

> > ### Comment · Reviewer_ByTS · 2026-07-20
> >
> > Thanks for all clarifications, fixes and new results in the paper! I am good with the new version!

---

### Review · Reviewer_ESPc · 2026-07-01

**Summary Of Contributions:**

This manuscript focuses on nonlinear time series modeling. The authors propose BPAR, which characterizes lag-specific nonlinear temporal dependencies through Bernstein polynomial basis functions. In addition, to capture linear associations, the authors further propose AR-BPAR, which improves the representational capacity of the model by introducing a linear AR branch. Experiments on both simulated and real-world datasets demonstrate strong empirical performance.

S1. Modeling nonlinear time series with polynomial basis functions is intuitive and reasonable, and it also improves the interpretability of the model.

S2. After fixing the Bernstein features, AR-BPAR can be formulated as a Bayesian linear regression model, which enables closed-form posterior inference, parameter credible intervals, and posterior predictive distributions. The mathematical formulation is intuitive and clear.

S3. The additionally introduced linear AR branch is also shown to be effective in the experiments.

S4. The experiments demonstrate the outstanding performance of the proposed method. In particular, Figure 5 shows a significant improvement over existing baselines.

Q1. Could the authors explain why, in some cases, MLP performs better than the existing baselines? More broadly, could the authors further clarify under what scenarios a simple model is already sufficient?

Q2. In addition, the authors mention several strong baselines in the introduction. Why are these methods not considered in the experiments? Q3. Since the Bernstein polynomial degree leads to a linear increase in the number of parameters, the authors should jointly consider the trade-off among polynomial degree, predictive performance, and model size.

Q4. For nonlinear time series, the assumption that \(\epsilon\) follows a Gaussian distribution may be overly restrictive and is often unlikely to hold in practice.

**Audience:**

Yes

**Audience Explanation:**

Time series forecasting, especially irregular time series forecasting, remains a central focus in machine learning research.

**Claims And Evidence:**

Yes

**Claims Explanation:**

The authors demonstrate the effectiveness of the proposed module through extensive experiments and analyses.

**Requested Changes:**

Please see Q1-Q4

---

> ### Author Response · Authors · 2026-07-15
> **Q1. When is a simple model sufficient?**
>
> *Could the authors explain why, in some cases, MLP performs better than the existing baselines? More broadly, could the authors further clarify under what scenarios a simple model is already sufficient?*
>
> **Response**: Firstly, the empirical study now spans datasets from areas namely dynamical systems in physics, ecology, hydrology, energy, environmental science, meteorology, transportation, astronomy and biology. See Appendix B.5 of the revised manuscript for the list of datasets considered. Section 3.2 of the revised manuscript also has details on these additions. This expanded list of datasets helps us better understand the differences in performance of the competing methods.
>
> The simpler models' advantage tracks the degree of nonlinearity present in the time series. It is seen, from our empirical studies, that a simple model suffices when the temporal dependence is close to linear. The expanded empirical study makes both directions visible in Table 11 in Appendix B of the revised manuscript. There, with highly nonlinear time series, the simpler models perform poorly. For instance, in that Table 11 under the Santa Fe time series, we see that ridge AR is $7.1\times$ worse than the best model. On the other hand, with the Beijing PM2.5 air pollution time series that is not highly nonlinear, all competing methods witness comparable performance. We dive deeper into this linear vs nonlinear testing in the newly added Appendix A.5.
>
> A key feature of the new AR-BPAR model is that it combines a linear AR branch with a nonlinear BPAR branch, and this helps capture varying degrees of linear/nonlinear temporal autocorrelations.

---

> ### Author Response · Authors · 2026-07-15
> **Q2. Why are the strong baselines from the introduction not in the experiments?**
>
> *In addition, the authors mention several strong baselines in the introduction. Why are these methods not considered in the experiments?*
>
> **Response**: We have now expanded the competing methods considered and the new additions involve both classical statistical methods and also modern deep learning based methods.
>
> Among the competing methods listed in Figure 7 (and Table 12 in Appendix B) of the revised manuscript, the newly added classical statistical suite includes techniques such as ridge-regularized autoregression (ridge AR) [10] and  exponential smoothing (ETS) [11]. Among more  modern deep learning based forecasters, we consider N-BEATS [15], DeepAR [18], a compact Transformer [19], and N-HiTS [1]. We also list capacity-matched MLP and LSTM variants, whose model sizes are enlarged so their trainable-parameter count matches or exceeds the size of the selected AR-BPAR configuration. In Tables 11-14 in Appendix B of the revised manuscript, the capacity-matched variants have "(matched)" written. Forecasting performance of all methods are evaluated using the same  rolling one-step ahead protocol.
>
> Table 11 in the revised manuscript reports, per dataset, held-out test RMSE relative to the best method on that dataset. Table 12 shows the resulting overall ranking of different methods in terms of forecasting performance. In these tables, "MLP (matched)" and "LSTM (matched)" denote the capacity-matched variants: the same MLP/LSTM architectures re-tuned with hidden sizes enlarged so that their trainable parameter count matches or exceeds the selected BPAR/AR-BPAR model configuration; see also  Appendix A.4 of the revised manuscript.  We report, in Appendix A.4, the criteria used to select the configurations of the competing MLP and LSTM methods, and also the competing linear time series models such as SARIMA/ARIMA.
>
> We see that across the 19 datasets, winners are spread over different methods. However, it must be noted that the new BPAR/AR-BPAR is consistently best-or-near-best in most cases. Table 12 in Appendix B of the revised manuscript reports the mean rank of the various methods along with the median relative test RMSE. Overall, we witness the strong performance of the AR-BPAR method.
>
> Finally, Table 14 in Appendix B of the revised manuscript compares the model sizes, including the newly added deep learning based methods. We see the advantage of the new AR-BPAR model which delivers strong forecasting performance along with a noticeably less model size when compared to the deep learning methods.
>
> ---
> **References**
>
> [1] Cristian Challu, Kin G Olivares, Boris N Oreshkin, Federico Garza Ramirez, Max Mergenthaler Canseco, and Artur Dubrawski. N-hits: Neural hierarchical interpolation for time series forecasting. In Proceedings of the AAAI .
>
> [10] Arthur E Hoerl and Robert W Kennard. Ridge regression: Biased estimation for nonorthogonal problems. Technometrics, 12(1):55–67, 1970.
>
> [11] Rob J. Hyndman and George Athanasopoulos. Forecasting: Principles and Practice. OTexts, Melbourne, Australia, 3rd edition, 2021.
>
> [15] Boris N. Oreshkin, Dmitri Carpov, Nicolas Chapados, and Yoshua Bengio. N-beats: Neural basis expansion analysis for interpretable time series forecasting. In International Conference on Learning Representations, 2020.
>
> [18] David Salinas, Valentin Flunkert, Jan Gasthaus, and Tim Januschowski. Deepar: Probabilistic forecasting with autoregressive recurrent networks, 2017.
>
> [19] Ashish Vaswani, Noam Shazeer, Niki Parmar, Jakob Uszkoreit, Llion Jones, Aidan N Gomez, Lukasz Kaiser, and Illia Polosukhin. Attention is all you need. In Advances in Neural Information Processing Systems, volume 30, 201.

---

> ### Author Response · Authors · 2026-07-15
> **Q3. Trade-off among degree, performance, and model size**
>
> *Since the Bernstein polynomial degree leads to a linear increase in the number of parameters, the authors should jointly consider the trade-off among polynomial degree, predictive performance, and model size.*
>
> **Response**:
> Note that we rely on the technique in Section 2.4 to select the optimal hyperparameter configuration. There, polynomial degree $m$ is one of the parameters considered in the search, but we also consider the prior distribution's $\tau$ and $\gamma$ parameters during that search. While a high polynomial degree may lead to better accuracy, the $\tau,\gamma$ parameters control overall model size.  Thus our approach here can be viewed as aiming for a balance between accuracy and overall model size.
>
> Table 10 in Appendix A.4 of the revised manuscript reports the selected Bernstein polynomial degree across the 19 real-world datasets. We witness this polynomial degree range from $m=5$ to $m=40$. As an illustration with a specific chaotic time series dataset (Santa Fe laser), we report sensitivity analysis with respect to the polynomial degree $m$. Figure 34 in Appendix B of the revised manuscript shows that the forecasting RMSE attains near optimality $m\approx 10$–$20$,  and plateaus beyond it. Figure 36 contains the corresponding sensitivity analysis plots for the Rössler time series dataset, another highly nonlinear time series. The observations made with these two representative datasets were seen to broadly hold true for the other 17 real-world datasets considered in this article.
>
> With the exact Bayesian inference method in Section 2.3.2 of the manuscript, the computation involves $K_{\theta}\times K_{\theta}$ sized matrix operations, and the associated cost grows polynomial with respect to the Bernstein polynomial degree $m$. Across the 19 real-world datasets considered in this article, the model size is seen to be $K_{\theta} \le 300$. The exact Bayesian inference method from Section 2.3.2 takes at most 10 milliseconds. Further, the complete hyperparameter search involving the technique from Section 2.4 takes a  median time of 9 seconds (maximum 31 seconds) per dataset on a single CPU core, with no GPU required. In comparison, the variational Bayesian inference route from Section 2.3.1 of the manuscript, estimated via backpropagation, is slower. A single model fit takes about 2 seconds on average, and the corresponding complete hyperparameter search takes a median time of approximately 2{,}060 seconds (maximum approximately 4{,}040 seconds) per dataset on the same single CPU core.
>
> Table 14 shows that even the smallest model  configuration among the neural network/deep learning competitors is larger than our AR-BPAR's model size, and this underscores the parsimony of the proposed model without compromising too much on prediction accuracy. Figure 33 shows the resulting accuracy–size frontier. There, we see that the proposed BPAR/AR-BPAR models reach top-tier accuracy at parameter counts that are orders of magnitude below the larger neural network/deep learning competitors. This discussion is added to Appendix B.4 of the revised manuscript.

---

> ### Author Response · Authors · 2026-07-15
> **Q4. Is the Gaussian error assumption too restrictive?**
>
> *For nonlinear time series, the assumption that $\epsilon$ follows a Gaussian distribution may be overly restrictive and is often unlikely to hold in practice.*
>
> **Response**: The posterior predictive distribution is already heavier-tailed than Gaussian. After integrating out the variance, the posterior predictive is Student-$t$ (see Eq. 24 of the manuscript), so the marginal forecast has heavier tails than a Gaussian and reflects scale uncertainty. To add empirical evidence to this result, we add Appendix B.1 to the revised manuscript.
>
> In Appendix B.1, we discuss  certain measures used for assessing the quality of prediction intervals obtained by applying the Bayesian procedure from Section 2.3.2 of the manuscript. First, we consider Prediction interval coverage probability (PICP) that measures the fraction of test points that the interval captures. A well-calibrated model has $PICP(\alpha)\approx 1-\alpha$; see [12] for details. Second, the mean relative prediction interval width ($MPIW_{rel}$) from  [12] which speaks to the sharpness of the interval. $PICP$ and $MPIW_{rel}$ must be read together: coverage is only meaningful at a stated sharpness, since an arbitrarily wide interval attains $PICP=1$ trivially. Third, the continuous ranked probability score (CRPS) that assesses the entire predictive distribution against the realized value; see [6] for details.
>
> Table 13 in Appendix B.1  reports, per dataset, the PICP at the coverage $90\%$ and $95\%$ levels, the relative MPIW, and the CRPS of AR-BPAR's closed-form Student-$t$ intervals.  Note that for BPAR/AR-BPAR models, these metrics evaluate the exact Student-$t$ posterior predictive intervals; see Section 2.3.2. Across the 19 real-world datasets, AR-BPAR's $90\%$ intervals attain a median empirical coverage of $0.92$.
>
> In that table, only  DeepAR's CRPS is displayed alongside for comparison because DeepAR is a competing method with a native predictive distribution at comparable point accuracy, and also is the strongest probabilistic baseline in the study (best CRPS on 13 of 19 datasets). Hence, it provides the most demanding like-for-like reference for BPAR/AR-BPAR's interval quality.  ARIMA/SARIMA analytic intervals are summarized in the table footer. DeepAR's intervals are computed as follows: the model is trained with a Gaussian likelihood on the network output, and its predictive distribution at each test step is obtained by ancestral sampling of full trajectories from the fitted model [18], as implemented in the `darts` package [8]; we draw $500$ sample paths per rolling one-step forecast, summarize them by their sample mean and standard deviation, form the $90\%$/$95\%$ intervals from the corresponding Gaussian quantiles, and evaluate the CRPS with the closed-form expression for a Gaussian predictive [6]. We see that BPAR/AR-BPAR attains the best CRPS at 6 of 19 datasets, but  delivers a performance that is highly competitive with DeepAR. It must be emphasized that the new BPAR/AR-BPAR model operates at a small fraction of the parameter count used by deep learning; see the newly added Table 14 in Appendix B for the exact counts.
>
> Finally, as an illustration, we consider two highly nonlinear time series datasets from our list (Santa Fe laser and Rössler) and plot the nominal and empirical coverage of the posterior predictive intervals; see Figure 32 in Appendix B.1 of the revised manuscript.
>
> ---
> **References**
>
> [6] Tilmann Gneiting and Adrian E Raftery. Strictly proper scoring rules, prediction, and estimation. Journal of the American Statistical Association, 102(477):359–378, 2007.
>
> [8] Julien Herzen, Francesco L"assig, Samuele Giuliano Piazzetta, Thomas Neuer, L'eo Tafti, Guillaume Raille, Tomas Van Pottelbergh, Marek Pasieka, Andrzej Skrodzki, Nicolas Huguenin, et al. Darts: User-friendly modern machi.
>
> [12] Abbas Khosravi, Saeid Nahavandi, Doug Creighton, and Amir F Atiya. Comprehensive review of neural network-based prediction intervals and new advances. IEEE Transactions on Neural Networks, 22(9):1341–1356, 2011.
>
> [18] David Salinas, Valentin Flunkert, Jan Gasthaus, and Tim Januschowski. Deepar: Probabilistic forecasting with autoregressive recurrent networks, 2017.